# m³C32 tRNA modification controls serine codon-biased mRNA translation, cell cycle, and DNA-damage response

Jia Cui[1,2,8], Erdem Sendinc[1,2,8], Qi Liu[1,2,3,4], Sujin Kim[1,2], Jaden Y. Fang[1,2] & Richard I. Gregory [1,2,5,6,7] ✉

The epitranscriptome includes a diversity of RNA modifications that influence gene expression. N3-methylcytidine (m³C) mainly occurs in the anticodon loop (position C32) of certain tRNAs yet its role is poorly understood. Here, using HAC-Seq, we report comprehensive METTL2A/2B-, METTL6-, and METTL2A/2B/6-dependent m³C profiles in human cells. METTL2A/2B modifies tRNA-arginine and tRNA-threonine members, whereas METTL6 modifies the tRNA-serine family. However, decreased m³C32 on tRNA-Ser-GCT isodecoders is only observed with combined METTL2A/2B/6 deletion. Ribo-Seq reveals altered translation of genes related to cell cycle and DNA repair pathways in METTL2A/2B/6-deficient cells, and these mRNAs are enriched in AGU codons that require tRNA-Ser-GCT for translation. These results, supported by reporter assays, help explain the observed altered cell cycle, slowed proliferation, and increased cisplatin sensitivity phenotypes of METTL2A/2B/6-deficient cells. Thus, we define METTL2A/2B/6-dependent methylomes and uncover a particular requirement of m³C32 tRNA modification for serine codon-biased mRNA translation of cell cycle, and DNA repair genes.

Chemical modification of RNA plays an important role in the post-transcriptional control of gene expression in normal cell homeostasis[1], development[2], and human disease[3]. According to the MODOMICS database, over 300 different chemical modifications on RNA have been identified[4]. Several of these marks are known to be dynamically installed and/or removed by specific RNA-modifying enzymes (writers and erasers, respectively)[3]. Recent technological advances[5] have helped reveal that certain RNA modifications play essential roles in controlling RNA stability, structure, localization, base-pairing interactions, interaction with RNA-binding proteins (RBPs), pre-mRNA splicing, mRNA polyadenylation, and mRNA translation[1]. For example, N⁶-methyladenosine (m⁶A), the most abundant and well-studied modification on mRNA, has been shown by numerous studies to influence gene expression via effects on mRNA stability and/or translation[6–8]. Aberrant m⁶A modification is involved in multiple human pathologies including cancer[9], cardiovascular disease[10], neurodegenerative disease[11], and diabetes[12]. Due to the oncogenic role of the m⁶A methyltransferase (MTase) METTL3, targeting the m⁶A epitranscriptome by pharmacological inhibition of enzymes controlling m⁶A homeostasis including METTL3 represents a promising strategy for treating acute myeloid leukemia (AML) and likely other cancer types[13,14].

tRNA is one of the most heavily modified RNA species in cells[15] and is responsible for decoding mRNA into corresponding amino acids via codon–anticodon interactions. The diverse chemical modifications of the anticodon and main body of tRNAs are of great importance for

[1]Stem Cell Program, Division of Hematology/Oncology, Boston Children's Hospital, Boston, MA 02115, USA. [2]Department of Biological Chemistry and Molecular Pharmacology, Harvard Medical School, Boston, MA 02115, USA. [3]Rice Research Institute, Guangdong Academy of Agricultural Sciences, 510640 Guangzhou, Guangdong Province, China. [4]Guangdong Key Laboratory of New Technology in Rice Breeding, 510640 Guangzhou, Guangdong Province, China. [5]Department of Pediatrics, Harvard Medical School, Boston, MA 02115, USA. [6]Harvard Initiative for RNA Medicine, Boston, MA 02115, USA. [7]Harvard Stem Cell Institute, Cambridge, MA 02138, USA. [8]These authors contributed equally: Jia Cui, Erdem Sendinc. ✉e-mail: rgregory@enders.tch.harvard.edu

tRNA integrity and function, thereby affecting protein synthesis[16]. For example, a variety of chemical modifications are found at the wobble position 34 of the tRNA anticodon to ensure accurate codon recognition and efficient decoding during mRNA translation[16,17]. Deficiency in 5-taurinomethyluridine (τm⁵U) modification at the wobble position of mitochondrial (mito-) tRNA-Leu-TTR has been reported as the primary cause for MELAS syndrome (mitochondrial myopathy, encephalopathy, lactic acidosis, and stroke-like episodes)[18]. Modifications within the tRNA-body regions (T-, D-, and variable loop) are usually involved in maintaining tRNA structure and stability. For example, $N^7$-methylguanosine ($m^7G$) at position 46 of the variable loop of cytoplasmic (cyto) tRNAs regulates normal stem cell differentiation and drives tumorigenesis via its control over tRNA stability[19,20]. Growing evidence supports that tRNA modifications and tRNA-modifying enzymes are involved in various human diseases, and targeting these enzymes could, therefore, offer new therapeutic opportunities[21]. However, the function of most of the various modification and editing events on different tRNAs remains poorly understood.

N3-methylcytidine ($m^3C$) is found on both cytoplasmic and mitochondrial tRNAs (cyto-RNAs and mito-RNAs) in human cells[22]. Unlike the initial reports that $m^3C$ is present only at position 32 of the anticodon loop of tRNA-Ser and tRNA-Thr in *S. cerevisiae*[23,24], human tRNAs are $m^3C$-modified at position 32 (C32) of tRNA-Ser, tRNA-Thr, two sets of tRNA-Arg isodecoders (tRNA-Arg-CCT, and tRNA-Arg-TCT), and also at C47d in the variable arm of tRNA-Ser and tRNA-Leu-CAG, as well as $m^3C20$ in the D-arm of tRNA-Met-CAT[22,25]. $m^3C$ is catalyzed by a family of SAM-dependent methyltransferases (MTases). In *S. cerevisiae* all $m^3C32$ on tRNA-Ser and tRNA-Thr depends on a single gene, *TRM140*[23,24]. However, in *S. pombe*, $m^3C32$ on tRNA-Ser or tRNA-Thr relies on one of two genes *trm140⁺* and *trm141⁺*, respectively[26]. Interestingly, humans have four known orthologs of yeast $m^3C$ MTases – METTL2A, METTL2B, METTL6, and METTL8. Human METTL8 was first identified as a potential mRNA $m^3C$ MTase[27]. However, recent data challenge this view since the predominant METTL8 isoform is localized in mitochondria, methylates $m^3C32$ of mito-tRNA-Thr/Ser-TCN and regulates mitochondrial mRNA translation[28,29]. Several studies have tried to identify the tRNA substrates of METTL2A, METTL2B, and METTL6 using in vitro methylation assays, primer extension assays, and/or mass spectrometry analysis[27,30–32]. However, a transcriptome-wide analysis of METTL2A/2B/6-mediated $m^3C$ tRNA methylomes in human cells is lacking. Moreover, it is still largely unknown how METTL2A/2B/6-mediated $m^3C$ methylation influences tRNA function in mRNA translation.

We recently developed Hydrazine-Aniline Cleavage sequencing (HAC-Seq), a sequencing technique to comprehensively profile $m^3C$ RNA modification throughout the transcriptome at single-nucleotide (nt) resolution[22]. Here, using HAC-Seq combined with $m^3C$ MTase-knockout experiments, we reveal METTL2A/2B-, METTL6-, and METTL2A/2B/6-dependent $m^3C$ methylomes in human cells. As expected, METTL2A/2B modifies C32 of tRNA-arginine and tRNA-threonine members, whereas METTL6 modifies C32 of the tRNA-serine family. Surprisingly, however, decreased $m^3C32$ on tRNA-Ser-GCT isodecoders was only observed with combined METTL2A/2B/6 deletion. We identify a conserved arginine residue (METTL2-R362 and METTL6-R259) at the C-terminal end of each protein that is essential for tRNA binding and further supports the overall substrate specificities of these $m^3C$ MTases. Diminished $m^3C32$ on cyto-tRNA by combined knockout of *METTL2A/2B/6* leads to altered mRNA translation. Specifically, loss of $m^3C32$ on tRNA-Ser-GCT causes decreased translation efficiency and increased ribosome stalling especially at serine AGU codons. We further find serine codon-biased alterations in mRNA translation of cell cycle and DNA damage genes associated with cyto-tRNA $m^3C32$ deficiency. As a result, *METTL2A/2B/6*-knockout cells show decreased cell growth with impaired cell-cycle progression and are more sensitive to the DNA-damaging drug cisplatin. Taken together, our study highlights the complexity of the $m^3C$ epitranscriptome in human cells and uncovers the molecular and cellular role of METTL2A/2B/6-mediated $m^3C32$ tRNA modification for normal mRNA translation, cell cycle progression, and response to DNA damage.

## Results

### MTase-dependent $m^3C$ tRNA methylomes

To define the role of known cytoplasmic $m^3C$ MTases in human cells, we first knocked out *METTL2A/2B*, *METTL6*, or *METTL2A/2B/6* using CRISPR/Cas9 technology in HEK293T cells (Supplementary Fig. 1a–d). METTL2A and METTL2B paralogs share 99% similarity in both their protein and mRNA coding sequences, and only one *Mettl2* gene exists in mice. Moreover, data has shown that while METTL2A and METTL2B both $m^3C$-modify tRNA, METTL2B exhibits only ~10% activity of METTL2A in vitro[32]. Therefore, we fully depleted human *METTL2* genes by simultaneously knocking out both *METTL2A* and *METTL2B*. We then analyzed the global $m^3C$ levels in total RNA or purified small RNA (smRNA <200nt) by mass spectrometry in the wild-type (WT) and knockout (KO) cells. We found that individual knockout of *METTL2A/2B* or *METTL6* each partially decreased $m^3C$ levels by ~30%. The combined knockout of three genes, *METTL2A/2B/6* (M2,6KO) resulted in a greater (~70%) yet incomplete loss of $m^3C$ RNA modification (Supplementary Fig. 1e–g).

We next performed HAC-Seq using these KO cell lines to uncover the $m^3C$ sites modified by METTL2A/2B and/or METTL6. HAC treatment induces site-specific cleavage at $m^3C$ on RNA[22] and the $m^3C$-modified sites are identified by calculating the cleavage ratio at each single nucleotide throughout the RNA (Fig. 1a). In HEK293T WT cells, $m^3C$-modified sites are found at C32 position on a subset of cytoplasmic tRNAs including tRNA-Arg-CCT/TCT, tRNA-Thr-AGT/CGT/TGT, and tRNA-Ser-AGA/CGA/TGA/GCT, as well as two mitochondrial tRNAs mito-tRNA-Ser-TGA, and mito-tRNA-Thr-TGT (Fig. 1a). The cleavage ratios of $m^3C32$ site on cytoplasmic tRNA-Ser and tRNA-Thr are above 95% indicating near stoichiometric modification of these tRNAs (Fig. 1a). Knockout of *METTL2A/2B* significantly reduced $m^3C32$ modification of tRNA-Arg and tRNA-Thr, whereas knockout of *METTL6* caused decreased m3C32 modification of tRNA-Ser (Fig. 1a–c). Upon full depletion of all three $m^3C$ MTases METTL2A/2B/6, $m^3C32$ modification is completely lost on the majority of $m^3C32$-modified cytoplasmic tRNAs (Fig. 1a). In contrast, our HAC-Seq data shows that $m^3C32$ on mito-tRNAs is not catalyzed by METTL2A/2B or METTL6 in human cells (Fig. 1a). Consistent with recent reports[28,29], we also found that METTL8 is the enzyme responsible for $m^3C32$ modification on mito-tRNAs, and it does not influence cyto-tRNA modification (data not shown). Furthermore, $m^3C$ was also found at C47d position on tRNAs with long variable arms including tRNA-Ser-AGA/CGA/TGA/GCT and tRNA-Leu-CAG; and at C20 position on tRNA-Met-CAT (Fig. 1a). However, these $m^3C$ modification sites are not affected by *METTL2A/2B* and/or *METTL6* knockout (Fig. 1a, d). The MTase(s) responsible for these $m^3C$ sites require further investigation.

Compared with other methods[27,30–32], HAC-Seq is a powerful tool to precisely study METTL2A/2B- and METTL6-mediated $m^3C$ modification of different tRNA isodecoders and isoacceptors. Overall, we found that METTL2 is primarily responsible for modifying a subset of cytoplasmic arginine and threonine tRNAs, whereas METTL6 is the major cytoplasmic serine tRNA-modifying enzyme. More specifically, we found that $m^3C32$ modification of two tRNA-Thr-CGT isodecoders (tRNA-Thr-CGT-1-1 and tRNA-Thr-CGT-3-1) was almost completely eliminated by *METTL2A/2B* KO (Fig. 1a). In contrast however, a very high cleavage rate of ~80% at $m^3C32$ was still observed on the other two tRNA-Thr-CGT isodecoders (tRNA-Thr-CGT-2-1 and tRNA-Thr-CGT-4-1) in *METTL2A/2B* and *METTL2A/2B*/6 KO cells (Fig. 1a), suggesting that unknown $m^3C$ MTase(s) are involved in the modification of certain tRNA-Thr-CGT isodecoders.

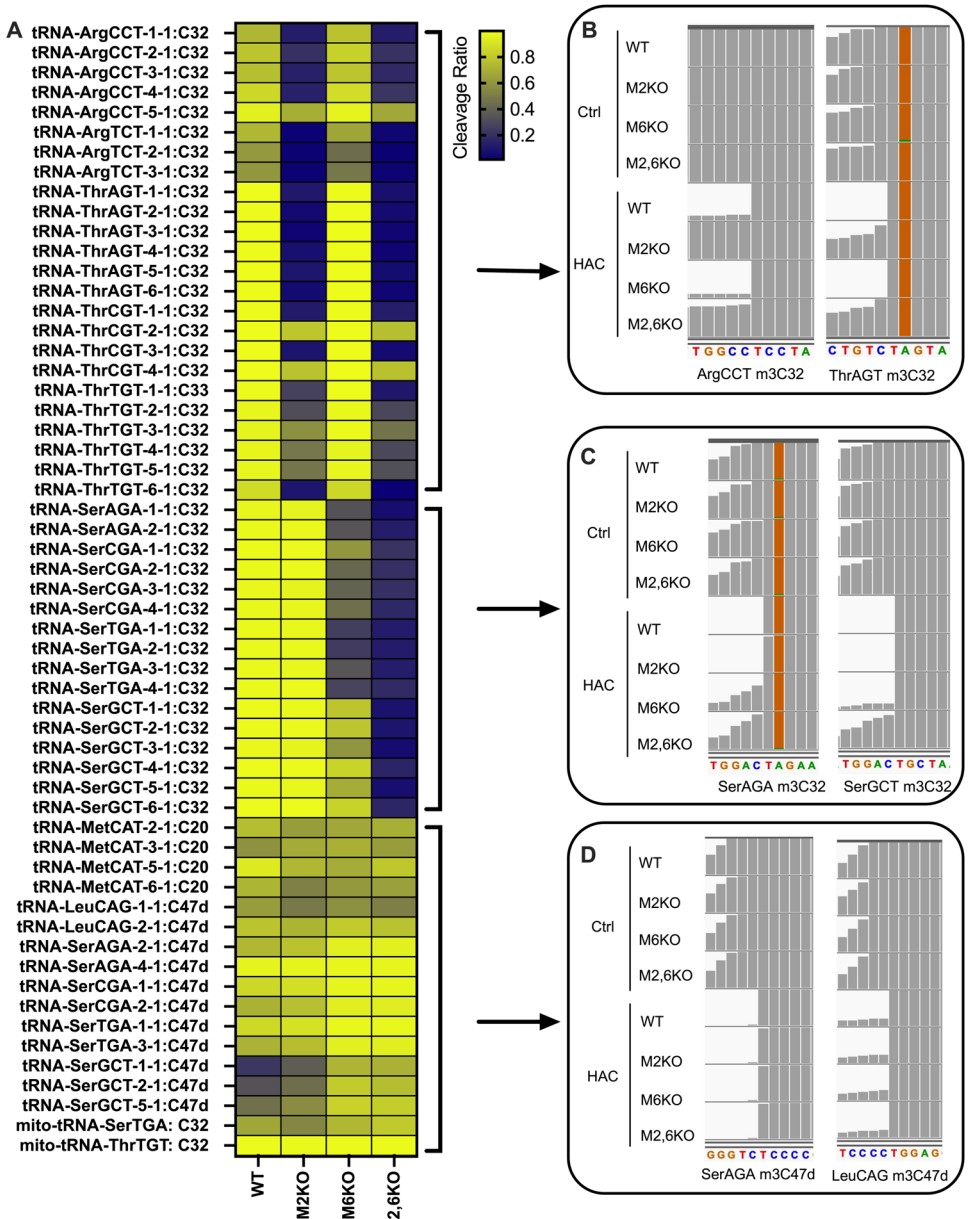

**Fig. 1 | HAC-seq analysis of METTL2A/2B- and METTL6-dependent m³C methylomes. a** Heat map showing the cleavage ratios of m³C modification sites detected by HAC-seq from WT, METTL2A/2B KO (M2KO), METTL6KO (M6KO), and MET-TL2A/2B/6 KO (M2,6KO) HEK293T cells. Data is shown as the mean value from two biological replicates. **b–d** A representative read alignments of tRNAs from untreated control (ctrl) and HAC-treated (HAC) groups of WT, M2KO, M6KO, and M2,6KO cells are shown in the Integrative Genomics Viewer (IGV). Representative plots of METTL2A/2B-prefered tRNAs (tRNA-Arg and tRNA-Thr) are shown in (**b**). Representative IGV plots of METTL6-prefered tRNAs (tRNA-Ser) are shown in **c** where m³C32 of tRNA-Ser-GCT is completely depleted only when METTL2A/2B/6 is knocked out. Orange labels shown in **b** and **c** represent sequencing mutation that reflects adenosine to inosine (A−I) editing at position A34. Representative IGV plots of METTL2A/2B/6-independent m³C modification sites are shown in (**d**). Source data are provided as a source data file.

Our HAC-seq data revealed that while METTL6 is generally required for m³C modification of tRNA-Ser isoacceptors, m³C32 on tRNA-Ser with A36 (tRNA-Ser-AGA/CGA/TGA) and tRNA-Ser with U36 (tRNA-Ser-GCT) are differentially impacted by *METTL6* knockout (Fig. 1). Single knockout of *METTL6* substantially decreased m³C32 on tRNA-Ser with A36 (Fig. 1a), whereas only a slight decrease in m³C32 on tRNA-Ser-GCT was observed (Fig. 1a, c). Notably, depletion of all three (*METTL2A/2B/6*) genes resulted in complete loss of m³C32 on tRNA-Ser-GCT (Fig. 1a, c). To verify the in vivo targets of these enzymes by an independent method, we carried out HPLC-MS/MS analysis of specific tRNAs isolated from WT and knockout cells. Consistent with HAC-seq data, we observed a complete loss of m³C on tRNA-Thr-AGT upon METTL2A/2B deletion, whereas loss of METTL6 did not impact m³C on

this tRNA, verifying specific in vivo activity of these enzymes (Fig. 2a). Strikingly, deletion of either *METTL2A/2B* or *METTL6* both decreased the m³C level on tRNA-Ser-GCT, and the combined loss of all three enzymes resulted a greater decrease in m³C level, indicating that METTL2A/2B and METTL6 are both required for m³C modification of this tRNA (Fig. 2a). Also of note, the residual m³C detected by mass spectrometry on tRNA-Ser-GCT in the M2,6 KO cells is likely due to the modification of C47d residue that is catalyzed by an unknown MTase (Figs. 1a and 2a). Taken together, these HAC-seq and mass spectrometry results indicate that while METTL2A/B and METTL6 enzymes generally display distinct specificities towards either tRNA-arginine and tRNA-threonine members, or the tRNA-serine family, respectively, these MTases (*METTL2A/2B/6*) are redundantly involved in the m³C32

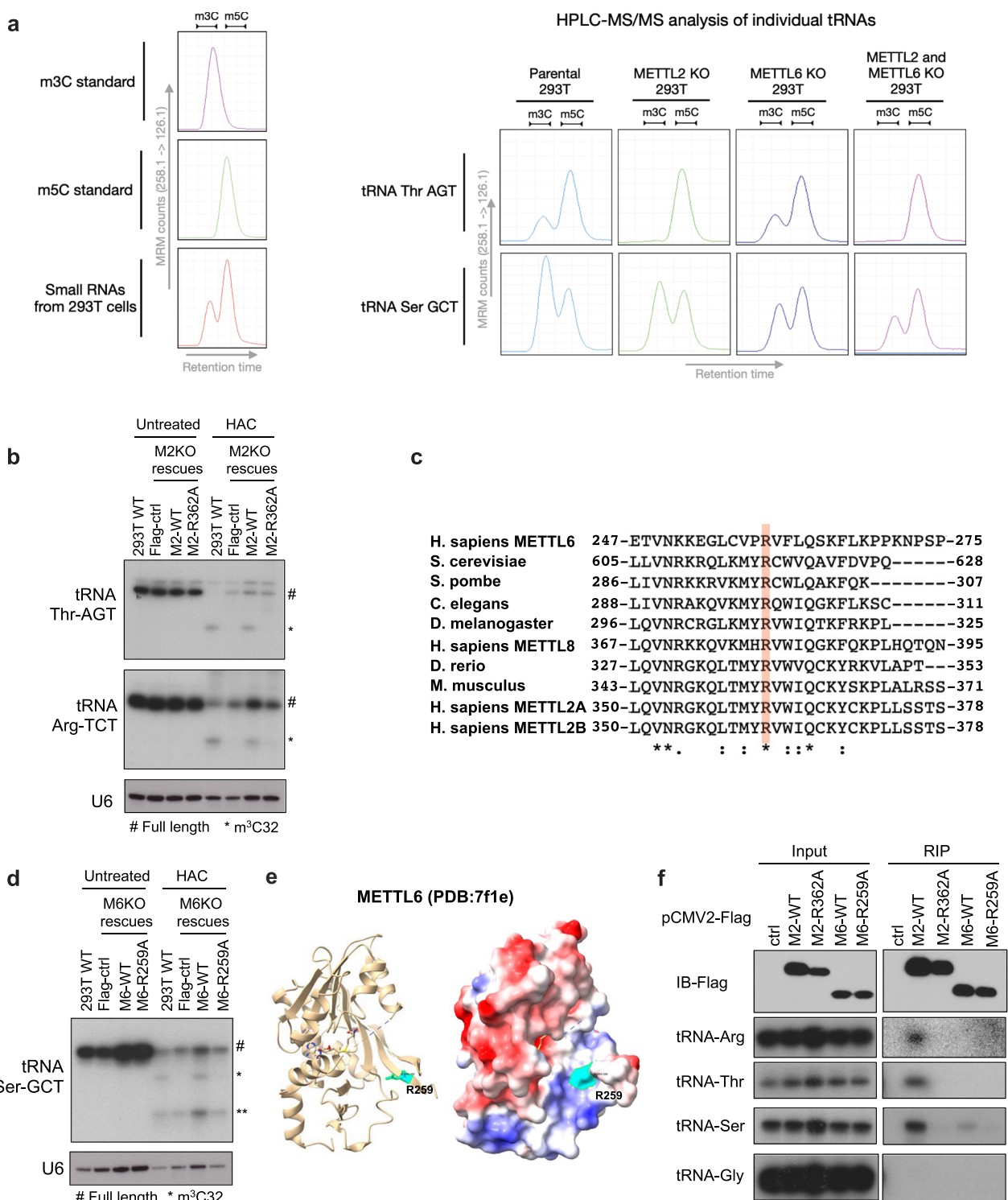

**Fig. 2 | A conserved C-terminal arginine residue required for METTL2A/2B/6 tRNA binding and methyltransferase activity. a** HPLC-MS/MS analysis of m³C and m⁵C standards (left). HPLC-MS/MS analysis of indicated tRNAs isolated from WT, *METTL2A/2B* KO, *METTL6* KO, or *METTL2A/2B/6* triple KO HEK293T cells (right). **b** Northern-blot detection of HAC-induced cleavage of tRNA at m³C showing re-expression of METTL2A (M2) but not the METTL2A-R362A mutant (M2-R362A) in METTL2A/B knockout (M2KO) cells can rescue the m³C modification on tRNA-Thr and tRNA-Arg. Blots are representative of repeated experiments. **c** Multiple sequence alignment of the C terminal end of m³C methyltransferases in different species. The conserved Arginine residue is highlighted in red. Accession numbers are NP_689609.2 (*H. sapiens* METTL6), NP_014882.4 (*S. cerevisiae*), NP_596587.2 (*S. pombe*), NP_001040827.1(*C. elegans*), NP_647636.3 (*D. melanogaster*),

NP_001308083.1 (*H. sapiens* METTL8), NP_001017902.1 (*D. rerio*), NP_766155.3 (*M. musculus*), NP_859076.3 (*H. sapiens* METTL2A), NP_060866.2 (*H. sapiens* METTL2B). **d** Northern-blot detection of HAC-induced cleavage of tRNA at m³C showing re-expression of METTL6 (M6) but not the METTL6-R259A mutant (M6-R259A) in METTL6 knockout (M6KO) cells can rescue the m³C modification on tRNA-Ser. Blots are representative of repeated experiments. **e** 3D structure of human METTL6 (PDB: 7f1e) shows the conserved R259 is localized in a positively charged groove next to the SAM binding pocket. **f** Northern blots of tRNAs interacting with WT and mutant proteins by RNA immunoprecipitation (RIP). M2-R362A and M6-R259A are not able to interact with their tRNA substrates. Blots are representative of repeated experiments. Source data are provided as a source data file.

modification of tRNA-Ser-GCT isodecoders. Altogether we report the first comprehensive METTL2A/2B- and/or METTL6-mediated tRNA m³C modification maps in human cells, uncover an unexpected complexity of m³C tRNA modification, and implicate currently unknown MTase(s) responsible for modification of certain m³C sites.

## tRNA binding and modification by METTL2/6

We next aimed to identify key amino acid residues in METTL2A/2B and METTL6 proteins required for their specific MTase activities. Previous research has shown that mutations in the conserved GxGxGx SAM-binding motif in METTL2 or METTL6 abolish their m³C methylation activity in vitro[27]. Additionally, it has been shown that METTL6 with mutations at N92 or F111 (corresponding N193 and F214 in METTL2A) cannot methylate RNA in vitro[31]. Therefore, we mutated these reported sites (G188A, G190A, G192A, G3A, dSAM, N193A, and F214A) as well as several other conserved sites (D213A, R291A, D292A, R302A, N312A, R321A, R354A, and R362A) in exogenously expressed METTL2A. G3A mutant replaces G188, G190, and G192 with three alanine. dSAM mutant is with full deletion of the GxGxGx motif in METTL2A. Since deletion of the C terminal end of TRM140 has been reported to abolish MTase activity in *S. cerevisiae*[24], we decided to also delete the C-terminal end (dC: deletion of aa 287–378) in human METTL2A. We first exogenously expressed these METTL2A mutants and wildtype (WT) proteins in HEK293T cells. We found that most of the METTL2A mutants did not express well and only METTL2A-N193A, -F214A, -R302A, -N312A, -R354A, and -R362A expressed METTL2A protein at levels comparable to METTL2A-WT (Supplementary Fig. 2a). We next performed rescue experiments by overexpressing a selection of the most robustly expressed METTL2A-mutants in HEK293T M2KO cells. Total RNAs from these cells were then analyzed by HAC-induced cleavage and Northern blot of individual m³C-modified tRNA. We found that only METTL2A-R362A failed to rescue the m³C32 modification on tRNA-Arg and tRNA-Thr in M2KO cells (Fig. 2b and Supplementary Fig. 2b). R362 is localized at the C-terminal end of METTL2A and is conserved in m³C MTases in different species (Fig. 2c). The corresponding mutant of METTL6 (METTL6-R259A) also expresses well but similarly was incapable of rescuing m³C on tRNA-Ser in METTL6 KO cells (Fig. 2d). METTL2A-R362 and METTL6-R259 are not located within the catalytic core of these MTases. The recently solved crystal structures of human METTL6 show that R259 is localized in a positively charged groove next to the SAM binding pocket (Fig. 2e), which could be the tRNA binding surface in METTL6[33,34]. In addition, since arginine is a positively charged amino acid that can hydrogen bond with RNA, we hypothesized that the reason why METTL2A-R362 and METTL6-R259 are essential for the m³C MTase activity is that these residues impact tRNA binding in cells. To test this, we performed RNA immunoprecipitation (RIP) assays to pull down tRNAs interacting with the mutant and wildtype METTL2A or METTL6 proteins in HEK293T cells (Fig. 2f). Northern blot analysis of tRNAs pulled down with the Flag-tagged proteins showed that mutations in METTL2A-R362 and METTL6-R259 abolished interaction with tRNA (Fig. 2f). Interestingly, while METTL6-WT protein was specifically bound to tRNA-Ser, METTL2A-WT protein interacts with all three m³C32-modified tRNA families (Arg, Thr, and Ser) in cells (Fig. 2f). These data further support our findings that endogenous METTL6 specifically binds and modifies tRNA-Ser, whereas METTL2A/2B binds to and modifies not only tRNA-Arg and tRNA-Thr, but also to certain tRNA-Ser members (Figs. 1, 2a, and f).

To explore the effect of m³C32 tRNA modification, we first examined tRNA expression changes upon M2KO, M6KO, or M2,6KO by analyzing smRNA-seq data. We found that knockout of *METTL2A/2B* and/or *METTL6* in HEK293T cells did not substantially alter the expression levels of tRNAs regardless of their m³C modification state (Supplementary Fig. 3a). Furthermore, METTL6 has been reported to interact with SARS (seryl-tRNA synthetase) in an RNA-dependent

manner[27,32]. It has been shown that SARS is essential for the m³C32 modification on tRNA-Ser-GCT in vitro, but its aminoacylation activity is not required[32]. Additionally, METTL2A and METTL2B have been identified to interact with DALRD3 whose C-terminal DALR domain is also found in RARS (arginyl-tRNA synthetase)[30]. DALRD3 lacks the tRNA aminoacylation activity, but its interaction with METTL2A/2B ensures efficient m³C formation on tRNA-Arg-CCT/TCT[30]. Therefore it is likely that tRNA aminoacylation is not a prerequisite for tRNA m³C32 modification, but whether m³C32 modification impacts tRNA aminoacylation is unknown. To explore this question, total RNA from HEK293T WT, M2KO, M6KO, and M2,6KO cells was extracted under acidic conditions to maintain the aminoacylated status of tRNAs. Equal amounts of total RNA were treated under alkaline conditions to deacylate tRNA. Then the charged and uncharged tRNAs were separated by acid-urea electrophoresis and detected by Northern blots. We found that the m³C32-modified tRNAs in HEK293T cells were 100% aminoacylated and the absence of m³C32 did not affect tRNA charging (Supplementary Fig. 3b).

## m³C32 is required for normal mRNA translation

Previous work has shown that *METTL2* KO in HEK293T cells does not affect global translation by polysome profiling[27], whereas knockout of *Mettl6* in mouse embryonic stem cells (mESCs) reportedly causes transcriptome-wide changes in mRNA abundance and ribosome occupancy[31]. The precise impact of m³C32 modification on tRNA function and mRNA translation remains unknown. Considering the relatively modest changes in global m³C levels we observed in either *METTL2A/2B* or *METTL6* KO cells (Supplementary Fig. 1e–g), as well as our evidence of functional redundancy between METTL2A/2B and METTL6 for certain Ser-tRNAs (Figs. 1 and 2), we utilized HEK293T WT and *METTL2A/2B/6* (M2,6KO) cells to explore the molecular and cellular role of cytoplasmic tRNA m³C32 modification.

Ribosome profiling (Ribo-Seq) analysis revealed altered translation efficiency (TE) of hundreds of mRNAs upon *METTL2A/2B/6* KO (Fig. 3a, b). Depletion of METTL2A/2B/6 resulted in 827 translation-upregulated genes (TE-Up) and 590 translation-downregulated genes (TE-Down) (1.5-fold change, adjusted $p < 0.05$) (Fig. 3b). We first compared the ribosome occupancy at each codon in cells with or without m³C32 tRNA modification (Fig. 3c). Although Thr, Arg (AGA, and AGG), and Ser codons are all decoded by m³C32-tRNA-dependent tRNAs, ribosome occupancy at these codons showed dramatically differential responses to tRNA m³C32 deficiency (Fig. 3c). Compared with WT, knockout of *METTL2A/2B/6* resulted in decreased ribosome A-site occupancy at Thr and Arg-AGR codons, while not affecting the control ribosome A+1 site. In contrast, however, increased ribosome A-site occupancy was observed at Ser codons (Fig. 3c), suggesting m³C32 on tRNA-Thr/Arg and tRNA-Ser have different effects on codon recognition during mRNA translation. Increased ribosome A-site occupancy indicates more ribosome stalling and decreased mRNA translation. Consistent with our ribosome occupancy analysis, TE-Down transcripts contained significantly more serine (Ser) codons than the TE-Up transcripts (Fig. 3d). However, there was little difference in the threonine (Thr) and arginine (Arg) codon content of these translationally regulated mRNAs (Fig. 3e). Altogether, we conclude that METTL2A/2B/6-mediated m³C32 tRNA modification is differentially required for mRNA translation of Arg/Thr compared to Ser codons, with a specific requirement of m³C32 for the efficient translation of serine codons.

## m³C32 controls serine codon-biased mRNA translation

Our ribosome occupancy analysis indicates that the most strongly impacted serine codon due to m³C32 deficiency is the near-cognate AGU codon that is decoded by tRNA-Ser-GCT (Fig. 3c). We next unbiasedly ranked the codon usage differences between the differentially translated mRNAs and found that serine AGU codon is one of the most highly enriched codons in the TE-down set of mRNAs (Fig. 4a,

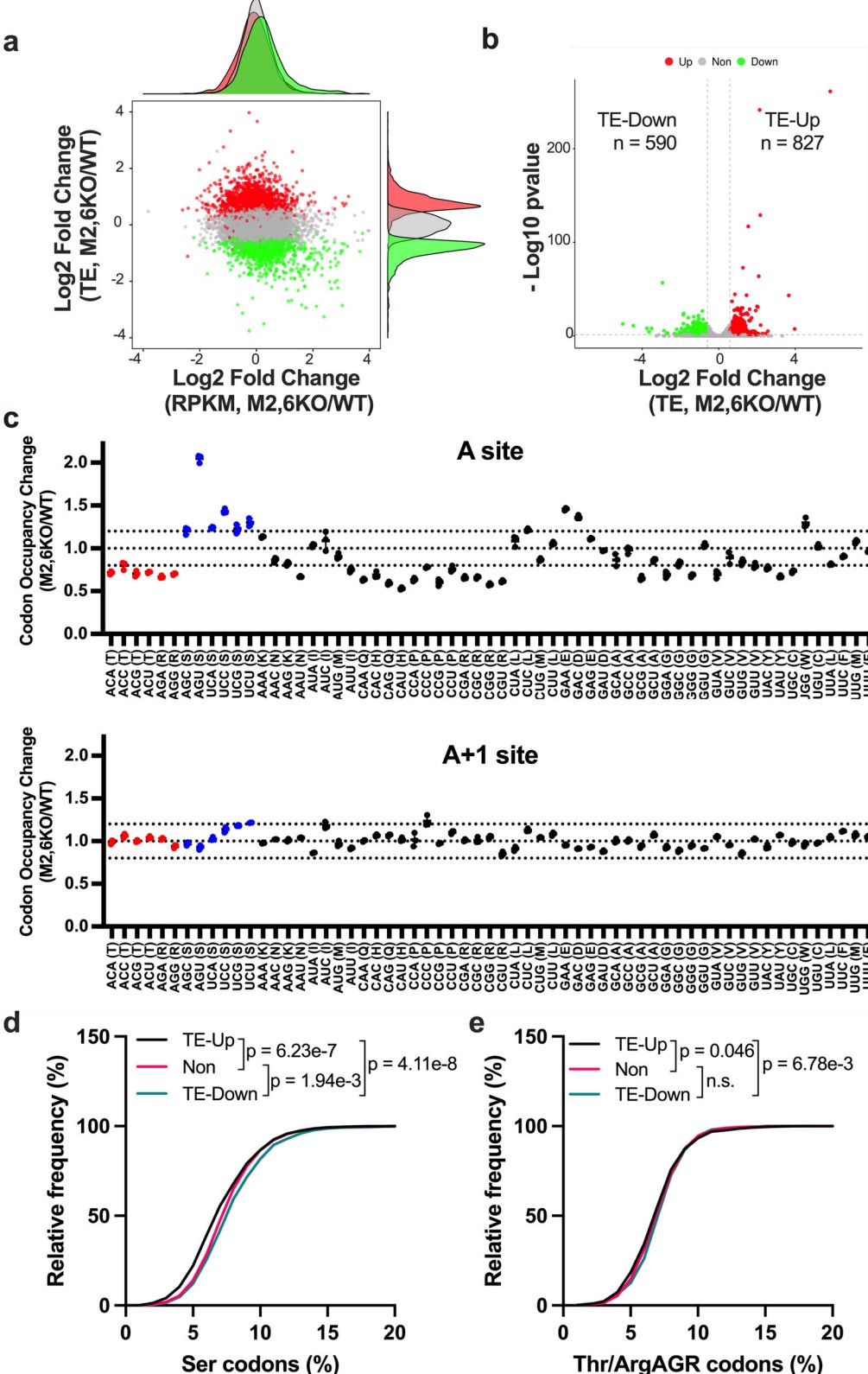

**Fig. 3 | METTL2A/2B/6-mediated m³C32 tRNA modification is required for normal mRNA translation. a** Scatterplot of mRNA-expression changes (*x*-axis) and translation-efficiency (TE) changes (*y*-axis) in *METTL2A/2B/6* knockout cells (M2,6KO) versus WT cells. TE was calculated by dividing the abundance of the ribosome-protected fragments (RPFs) by transcript abundance from the RNA-seq input. **b** Volcano plots showing differentially translated genes (1.5-fold change, adjusted *p* < 0.05). TE-Up: translationally upregulated genes; TE-Down:

translationally downregulated genes. **c** Ribosome occupancy at individual codons at A sites and A + 1 sites. Red: codons decoded by type I m³C32-modified tRNAs (tRNA-Arg/Thr). Blue: codons (Ser) decoded by type II m³C32-modified tRNAs (tRNA-Ser). Data are shown as mean ± SD of *n* = 3 biological replicates. **d**, **e** Cumulative frequency analysis of Ser (**d**) and ArgAGR/Thr (**e**) codon frequencies in TE-Up and TE-Down genes. *p* values were calculated using the two-sided Mann–Whitney test. Source data are provided as a source data file.

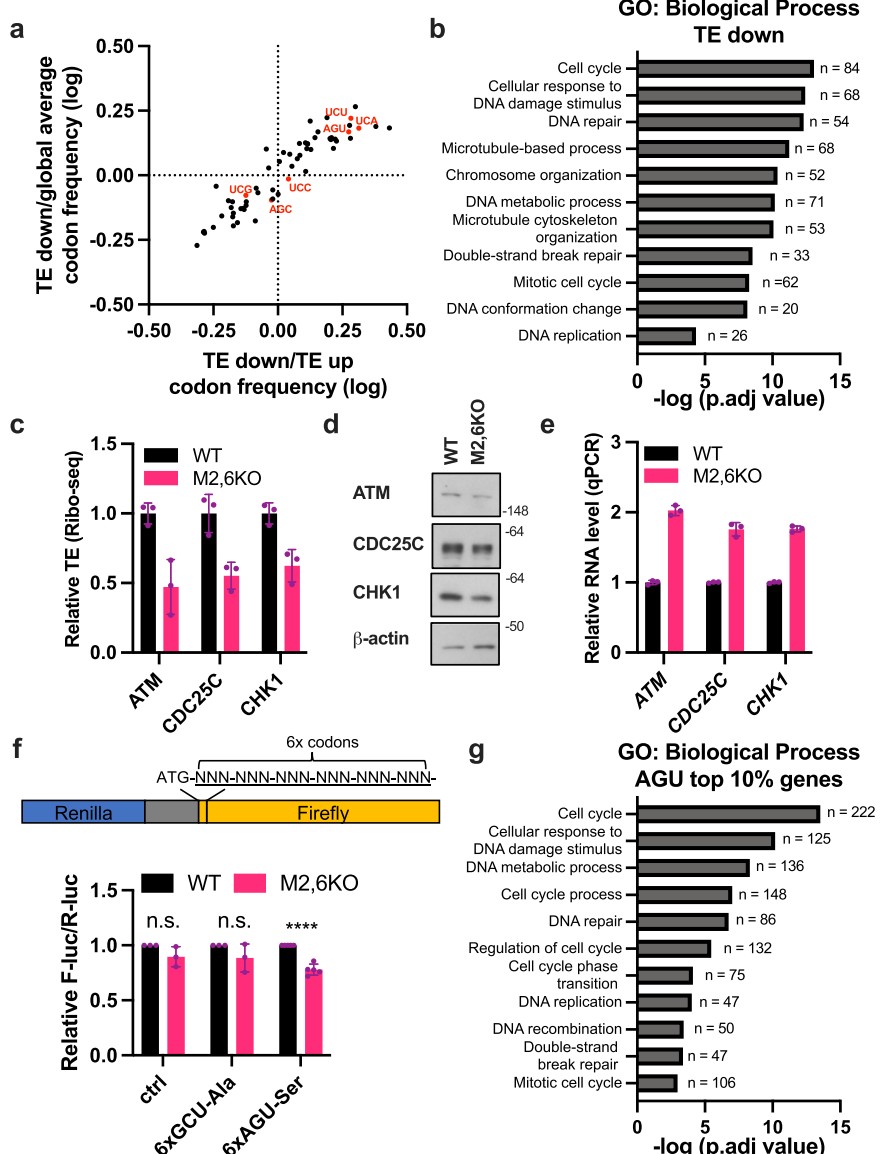

**Fig. 4 | m³C32 tRNA modification promotes serine codon-biased mRNA translation of cell cycle and DNA repair genes. a** Scatterplot of codon frequency changes in the differential translated genes (TE-Down vs. TE-Up and TE-Down vs. Global average). Serine codons are highlighted in red. **b** Gene ontology analysis of TE-Down genes in biological process. **c** TE changes of cell-cycle and DNA-repair regulators from Ribo-seq analysis. Data are shown as mean ± SD, $n = 3$ biological replicates. **d**, **e** protein and mRNA level changes of TE-down cell-cycle and DNA-repair regulators were validated by western blots and qPCR, respectively. Data is shown as mean ± SD, $n = 3$ technical replicates. **f** Dual-luciferase reporter assays showing m³C32 deficiency in M2,6KO cells decreased protein decoding of serine AGU codons. No-insertion (ctrl) and 6xGCU-Ala plasmids were included as controls. Data are shown in as mean ± SD, $n = 3$ biological replicates. The statistical significance was determined by unpaired two-tailed Student *t*-test where n.s. not significant; ****$p < 0.0001$ ($p = 0.000009$). **g** Gene ontology analysis of genes enriched with AGU codons in biological process. Source data are provided as a source data file.

Supplementary Fig. 4). Interestingly, while other serine codons (including UCA, and UCU) were also found to be significantly enriched in the TE-down mRNAs, other serine codons (i.e. AGC, UCC, and UCG) are not differentially distributed between the mRNAs with increased or decreased translation efficiency, thereby implying a specific requirement of m³C32 for the decoding of certain serine codons including AGU and involvement in codon-biased translation of sets of mRNAs that are enriched in these codons.

Gene ontology analysis of the Ribo-Seq dataset revealed that the set of mRNAs with decreased translation efficiency (TE) in the absence of cytoplasmic tRNA m³C32 (TE-down), are significantly associated with cell cycle and DNA repair pathways (Fig. 4b). Since tRNA decoding efficiency at serine AGU was identified as the most prominently impacted codon upon loss of m³C32 tRNA modification, we

hypothesized that m³C32-mediated translational control at serine AGU codons largely contributes to this enrichment of mRNAs involved in cell cycle and DNA repair. In support of this, our Ribo-seq analysis uncovered that mRNAs encoding known cell-cycle and DNA-repair regulators (ATM, CDC25C, and CHK1), are translationally down-regulated in *METTL2A/2B/6* KO cells and are amongst the set of mRNAs that are most highly enriched in AGU codons (Fig. 4c). In agreement, western blotting and qPCR analysis showed that the protein levels of these regulators were decreased (Fig. 4d) while their mRNA levels were increased in METTL2A/2B/6 KO cells compared to control cells (Fig. 4e).

Next, to directly test the functional requirement m³C2 tRNA modification for efficient mRNA translation, we designed and utilized a dual-luciferase reporter system where six tandem repeats of either

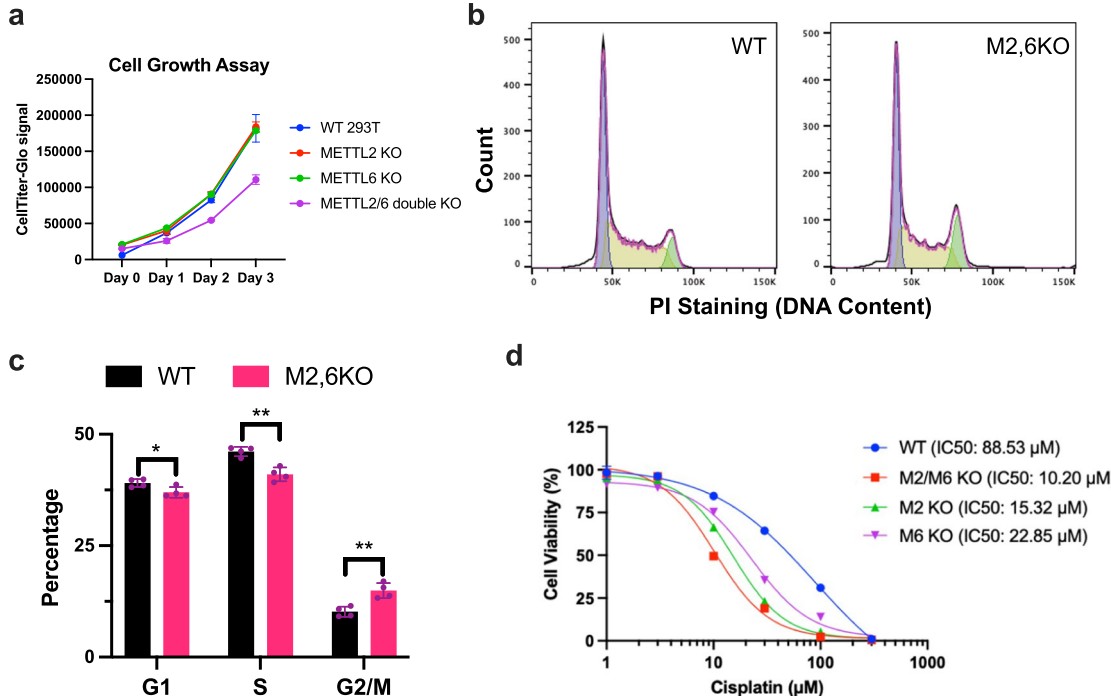

**Fig. 5 | METTL2A/2B/6 deficiency impairs cell cycle and response to DNA damage. a** Cell proliferation analysis of HEK293T WT, *METTL2A/2B*, *METTL6*, and *METTL2A/2B/6* triple KO cells. Data are shown as mean ± SD, *n* = 3 biological replicates **b**, **c** Cell cycle analysis of HEK293T WT and *M2,6KO* cells. Quantifications of cell populations in different cell cycle phases are shown as mean ± SD, *n* = 4 biological replicates. The statistical significance was determined by unpaired Student *t*-test where \**p* < 0.05; \*\**p* < 0.01. (unpaired two-tailed Student *t*-test *p* values for G1 = 0.03, S = 0.002, G2/M = 0.004). **d** Cell viability analysis of HEK293T WT, *METTL2A/2B*, *METTL6*, and *METTL2A/2B/6* triple KO cells with cisplatin treatment. Data are shown as mean ± SD, *n* = 3 biological replicates. Source data are provided as a source data file.

AGU codon or GCU codon (negative control), were inserted downstream of the ATG start codon of the firefly luciferase gene (Fig. 4f). We found that m³C32 deficiency in *METTL2A/2B/6* KO cells resulted in significantly decreased expression of the luciferase reporter mRNA containing AGU codons (Fig. 4f) thereby further supporting a specific requirement for METTL2A/2B/6-mediated m³C modification of tRNA-Ser-GCT for efficient decoding of mRNAs containing AGU Ser codons. Interestingly, an unbiased codon distribution analysis of all human mRNAs revealed that mRNAs encoding cell cycle and DNA damage response genes are highly enriched in AGU codons (Fig. 4g).

### m³C32 promotes cell-cycle progression and DNA repair

Since our molecular characterization showed the requirement for METTL2A/2B/6-mediated m³C modification of tRNA-Ser-GCT for efficient decoding of mRNAs containing AGU Ser codons and the enrichment of AGU codons in genes involved in cell cycle regulation and response to DNA damage, we next explored cellular phenotypes in m³C-deficient cells. We found that *METTL2A/2B/6* KO cells with tRNA m³C32 deficiency proliferated more slowly than WT or individual *METTL2A/2B* or *METTL6* KO cells (Fig. 5a) and showed altered cell cycle progression with an impaired G2/M transition in Fig. 5b, c and a. Similarly, while individual deletion of *METTL2* or *METTL6* had a modest effect on growth of MCF7 human breast cancer cells, the further siRNA-mediated depletion of either METTL6 in the *METTL2* KO cells, or knockdown of METTL2 in the *METTL6* KO cells strongly suppressed cell proliferation (Supplementary Fig. 5a–c). Furthermore, *METTL2A/2B/6* KO cells were more sensitive to the DNA-damaging drug cisplatin with a lower IC50 compared with WT cells or individual *METTL2A/2B* or *METTL6* KO cells (Fig. 5d).

### Discussion

This study applied HAC-Seq and genetic knockout approaches to comprehensively identify METTL2A/2B- and METTL6-mediated m³C

epitranscriptomes in human cells. We furthermore utilized Ribo-Seq to examine the consequences of METTL2A/2B/6-deficiency on mRNA translation and identified a particular requirement of m³C32 tRNA modification for efficient translation of cell cycle, and DNA repair genes that are enriched in AGU codons and require m³C-modified tRNA-Ser-GCT for normal translation. Accordingly, we found MET-TL2A/2B/6-deficient cells have slowed growth, impaired cell cycle progression, and increased sensitivity to DNA damage (Fig. 6).

Overall, we found that METTL2A/2B is primarily responsible for modifying a subset of cytoplasmic arginine and threonine tRNAs, whereas METTL6 is the major cytoplasmic serine tRNA-modifying enzyme at position C32. These findings agree well with the known roles of METTL2 and METTL6 in shaping the m³C tRNA methylome. Other m³C sites within the tRNA body regions (m³C47d and m³C20) were unaffected by METTL2A/2B/6-deletion and the identity of the MTase(s) responsible is unknown. The substrate recognition mechanism by human m³C MTases is largely unexplored. tRNA-Arg and tRNA-Thr both belong to the type-I tRNAs. They contain m³C32 and N6-Threonylcarbamoyladenosine A37 (t⁶A37) modifications in the anticodon loop. In vitro, methylation assays show that METTL2A alone methylates tRNA-Thr in a t⁶A37-dependent manner[32]. However, for tRNA-Arg, t⁶A37 is not sufficient for METTL2A-mediated m³C methylation in vitro[32], suggesting the requirement of other factors.

In contrast to tRNA-Arg/Thr, tRNA-Ser belongs to the type-II tRNA containing an additional hairpin in the variable arm. There are two types of tRNA-Ser in cells−tRNA-Ser with A36 (tRNA-Ser-AGA/CGA/TGA) and tRNA-Ser with U36 (tRNA-Ser-GCT). Compared with other methods, HAC-Seq is a powerful tool to precisely study METTL2A/2B- and METTL6-mediated m³C modification of different tRNA iso-decoders and isoacceptors and can, therefore, comprehensively reveal METTL2-, METTL6-, and METTL2 + 6-dependent m³C profiles that might have been missed in previous work. Our HAC-seq data

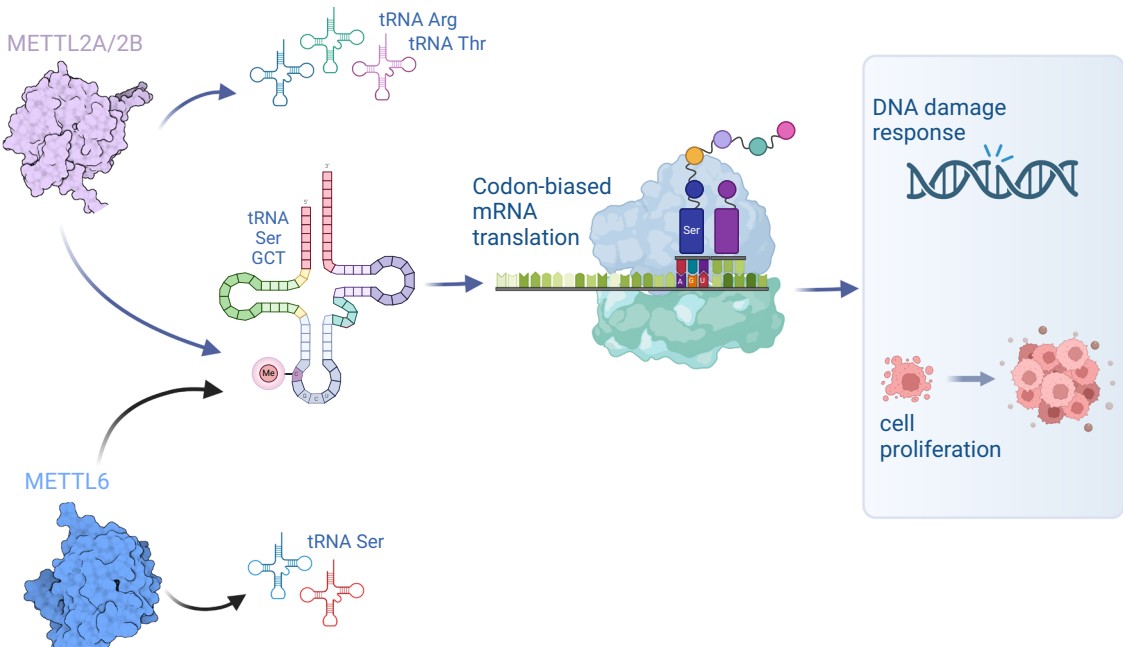

**Fig. 6 | m³C32 tRNA modification controls serine codon-biased mRNA translation, cell cycle, and DNA-damage response.** METTL2A/2B is generally responsible for m³C32 modification of tRNA-Arginine and tRNA-Threonine families, whereas METTL6 specifically modifies the tRNA-Serine family. However, METTL2A/2B and METTL6 display functional redundancy for the m³C32 modification of tRNA-Ser-GCT isodecoders in human cells. METTL2A/2B/6 are particularly required for the efficient translation of AGU codons that are decoded by tRNA-Ser-GCT. A subset of mRNAs encoding cell-cycle and DNA repair proteins are enriched in AGU codons, and METTL2A/2B/6-deficiency causes decreased expression of these genes and hence decreased proliferation, impaired cell cycle progression, and increased sensitivity to DNA-damage. This figure is created with BioRender.com released under a Creative Commons Attribution-NonCommercial-NoDerivs 4.0 International license.

uncovered that knockout of *METTL6* alone induces a much more profound reduction of m³C32 methylation on tRNA-Ser with A36 than that on tRNA-Ser with U36. Unexpectedly, we found that only knockout of all three MTases *METTL2A/2B/6* completely abolished the m³C32 methylation on all tRNA-Ser species. This suggests that the C32 positions of tRNA-Ser with A36 or U36 are differently dependent on METTL2A/2B and METTL6. The interaction between m³C MTase and Ser-tRNA synthetase (e.g., TRM140-Ses1 in *S. cerevisiae* and METTL6-SARS in humans) has been shown as a key determinant for m³C32 modification on tRNA-Ser[32,35]. Interestingly, TRM140-Ses1 interaction stimulates less efficient m³C32 modification on tRNA-Ser-GCT compared with tRNA-Ser-CGA/TGA in vitro[35]. Moreover, all tRNA-Ser species with A36 are N6-isopentenyladenosine A37 (i⁶A37)-modified while tRNA-Ser with U36 is t⁶A37-modified like the METTL2A/2B-preferred substrates tRNA-Arg and tRNA-Thr. In *S. cerevisiae*, i⁶A37 and t⁶A37 are important for m³C32 biogenesis on all tRNA-Ser species[35]. Structural studies have shown that i⁶A37 destabilizes whereas t⁶A37 stabilizes the tRNA anticodon loop structure[36,37], implying different structural conformations between these two types of tRNA-Ser species might help explain the differential dependency of serine tRNA isoacceptors on METTL2A/2B and/or METTL6 for m³C32 modification. Of note, tRNA-Ser-GCT was not found to be a substrate for METTL2 enzyme in reconstituted MTase assays in vitro[32], suggesting that additional cofactors and/or tRNA modifications can alter the METTL2 substrate determinants in vivo, and could help explain the functional redundancy we identified between METTL2 and METTL6 for m³C modification of tRNA-Ser-GCT in cells. We further speculate that the relative contribution of METTL2 and METTL6 to the m³C32 modification of tRNA-Ser-GCT might vary in different cell types due to the relative expression/activity of these different m³C MTases.

In addition, we report that mutation in the conserved arginine residue (METTL2A-R362 and METTL6-R259) at the C terminal end of these MTases disrupts their interaction with tRNA. We also find that METTL2A can interact with all m³C32-modified tRNAs including tRNA-Ser species. This could help explain how METTL2A/2B is capable of methylating tRNA-Ser-GCT in the absence of METTL6 in cells. Protein structure prediction of full-length METTL2A/2B and METTL6 by AlphaFold shows high folding similarity in their enzymatic core[38]. Nevertheless, METTL2A/2B contains an additional disordered region (aa105-173) localized on the opposite side of the putative tRNA binding surface. It is likely that this disordered region in METTL2A/2B plays a role in modulating tRNA m³C methylation specificity and efficiency. Taken together, tRNA sequence, folding, additional tRNA chemical modifications, and MTase cofactors may cooperatively determine the specific tRNA substrate recognition by different m³C32 modifying enzymes. The 3D structures of m³C MTase(s) in complex with different tRNA isoacceptors and/or protein cofactors would provide more information on the specific tRNA substrate recognition mechanisms.

By further characterizing the effects of m³C32 modification on cyto-tRNA functions, we identified that tRNA m³C32 deficiency did not affect tRNA expression or aminoacylation. Considering this, together with reports that m³C32 optimizes mito-tRNA structure[28,39], we speculate that m³C is required for optimal cyto-tRNA conformation. Position 32 is located just below the tRNA anticodon stem. It tends to form a non-Watson-Crick base pair with position 38 to adopt the U-turn structure in the tRNA anticodon loop for ribosomal binding[40]. All m³C32-modified tRNA in humans contain an A38. Although there is no structural evidence showing direct pairing between m³C32 and A38 or any surrounding nucleotide in a native tRNA, thermal denaturing studies using synthesized modified RNA show that m³C significantly decreases base pairing specificity between C-G pair and other mismatched pairs[41]. Furthermore, A37 in all m³C32-containing tRNA is modified as i⁶A or t⁶A. Molecular dynamics (MD) simulations reveal that i⁶A and t⁶A modifications impact the conformational flexibility of anticodon nucleotides[42]. Additional chemical modifications on the

anticodon loop may induce more significant changes in tRNA folding. Therefore, the chemical environment around C32-A37-A38 may play a critical role in optimizing the structural conformations of the tRNA anticodon loop, thereby fine-tuning mRNA translation in human cells.

We next characterized the impact of m³C32 on mRNA translation using Ribo-Seq and mRNA reporter genes. We found that the absence of m³C32 on tRNA-Ser, which belongs to type-II tRNAs results in increased ribosome occupancy and decreased mRNA translation. In contrast, the absence of m³C32 on tRNA-Arg/Thr, a group of type I tRNAs, leads to decreased ribosome occupancy. Translation of the near-cognate serine AGU codon is the most impacted by METTL2A/2B/ 6 -deficiency. Since the translation of AGU codons but not cognate AGC codons was significantly different between the translationally up versus downregulated mRNAs in the METTL2/6-deficient cells (Supplementary Fig. 4), we speculate that m³C modification of tRNA-Ser-GCT isodecoders is especially important in the context of a G:U wobble for normal translation. Future tRNA structural and functional studies should help understand the precise impact of m³C32 in normal mRNA translation. Loss of m³C32 translationally downregulated numerous regulators responsible for cell cycle progression and DNA repair. Since AGU-rich genes are enriched in cell-cycle and DNA-repair pathways, m³C32-mediated translational changes on AGU-rich genes could in part explain the cellular phenotypes we observed in the m³C32-deficiency cells (Fig. 6) and provide an additional example of how RNA modifications can contribute to codon-biased mRNA translation mechanisms for the coordinated expression control of functionally-related gene subsets[20,43–46].

Phenotypically, we found that depletion of m³C32 on tRNA by knocking out all three human m³C32 MTases METTL2A/2B/6 significantly reduced cell proliferation and hindered G2-M cell cycle transition, compromised DNA-damage response in HEK293T cells. Interestingly, expression of METTL2A, METTL2B, and METTL6 mRNA is elevated in most tumor types compared with corresponding normal tissues and is associated with poor prognosis[47]. High levels of MET-TL2A mRNA expression are associated with shortened overall survival in breast invasive carcinoma (BRCA) patients and associated with high-grade tumors[47]. Gene Ontology analysis of the set of mRNAs enriched in TCGA datasets based on METTL2A expression showed that several cell cycle-related pathways were enriched, such as chromosome segregation, and DNA biosynthetic process. These results indicate that BRCA cells with high METTL2A expression could be active in proliferation and DNA damage response pathways[47]. Another cancer genomics study identified METTL6 as one of eight essential genes highly amplified in patients with luminal-type breast cancer[48]. Down-regulation of METTL6 has been shown to inhibit liver cancer cell proliferation and invasion[49]. Our HAC-seq data from HEK293T cells and MCF7 cells[22] reports that m³C32 methylation is almost saturated especially at tRNA-Thr and tRNA-Ser species. Therefore, detailed investigations are required to further characterize why m³C MTases are overexpressed/amplified in multiple cancer types and how such overexpression contributes to cancer progression. Nevertheless, manipulation of m³C32 modification levels on tRNA by directly targeting m³C32 chemical marks or targeting m³C MTases may be a promising strategy for anti-cancer therapeutics in the future. DNA-damaging agents are widely used in the clinic for treating both solid and hematological cancers, yet resistance to DNA-damaging drugs is also one of the biggest challenges in chemotherapy. Our data suggests that combined targeting of both DNA repair pathways and m³C MTases may sensitize cancer cells to chemotherapy. In summary, m³C tRNA epitranscriptomics and its role in codon-biased translation is far more complicated than we expected. Although recent studies are beginning to uncover the molecular and cellular function of the m³C tRNA epitranscriptome[50], more efforts are needed to fully understand the m³C tRNA epitranscriptome and its role in normal cell homeostasis, development, and diseases.

## Methods

### Cell culture

HEK293T cells were cultured in DMEM high glucose medium supplemented with 10% fetal bovine serum, 2.05 mM L-glutamine, 1 mM sodium pyruvate, 100 U/ml each of penicillin and streptomycin in a humidified atmosphere with 5% $CO_2$ at 37 °C.

### Plasmid construction and transfection

For METTL2A or METTL6 overexpression, full-length cDNAs of MET-TL2A or METTL6 were PCR amplified from HEK293T cells with primers containing NotI and BglII restriction sites. The PCR products were then gel purified. pCMV2-Flag-emptry vector and the PCR inserts were digested with the corresponding registration enzymes and gel-purified. Next, the digested insert was ligated into the cut pCMV2-Flag-empty vector using T4 DNA ligase (NEB, #M0202). All pCMV2-Flag-METTL2A and pCMV2-Flag-METTL6 mutation constructs were generated by using a Q5 site-directed mutagenesis kit (NEB, #E0554) except pCMV2-Flag-METTL2A-dC. pCMV2-Flag-METTL2A-dC was generated by subcloning the cDNA of METTL2A with a deletion of amino acids 287-378 into the pCMV2-Flag-empty vector. For the construction of METTL2A/2B and/or METTL6 sgRNA vectors for CRISPR-Cas9 knockout, sgRNAs were designed using the CRISPR design website (http://crispr.mit.edu/). sgRNA specificity was checked by using the Cas-OFFinder website (http://www.rgenome.netcas-offinder/). For METTL2A/2B KO, sgRNAs were designed to target both METTL2A and METTL2B. Sense and antisense sgRNA oligos were annealed and ligated with BbsI-digested pX459 vector. All primer sequences used for plasmid construction are included in Table S1. Plasmid transfection was performed by using Lipofectamine 2000 (Thermo-Fisher, #11668019). Cells were harvested after 24 or 48 h transfection.

### Generation of CRISPR-Cas9 knockout cells and siRNA knockdown

HEK293T cells were transfected with a pair of pX459 vectors containing sgRNAs for KO cells or pX459 empty vectors for WT cells. sgRNA sequences were listed in Table S1. Cells were selected with puromycin (2.5 μg/mL) for 24 h and then switched to a complete medium without puromycin for screening single clones. Genomic DNA from single-cell colonies was extracted by using PureLink™ Genomic DNA Mini Kit (Invitrogen, #K182002). Knockout efficiency was validated by PCR of genomic DNA around the CRISPR-Cas9 targeting regions followed by sanger sequencing and by RT-qPCR of *METTL2A/2B* or *METTL6* mRNAs. PCR was performed by using Q5® High-Fidelity 2x Master Mix (NEB, #M0492S). PCR products were separated by 2% agarose gels. The following PCR validation primers were used (5' to 3'): hMETTL2A/B KO validation (F): GTCTTCTGAAAGAGGGCGTG; hMETTL2A/B KO validation (R): CCTAGGAGTTTCTCTCCCCTTG; hMETTL6 KO validation (F): AAATACTGGGACACATTTTACAAGA; hMETTL6 KO validation (R): TGCACAGAAGTACACTGAATGCT. For siRNA-mediated knockdown the following siRNAs were used: Integrated DNA Technologies (IDT) Cat #: 51-01-14-04; 20003606; CD.Ri.476720.13.2; hs.Ri.METTL6.13.1.

### Total RNA extraction and RT-qPCR

Total RNA was extracted by TRIzol™ reagent (Invitrogen, #15596026) following the manufacturer's protocol. 1 μg of total RNA was reversed transcribed to cDNA by using SuperScript™ III (Invitrogen, #18080093) and random hexamers (Invitrogen, #N8080127). qPCR reactions were carried out using Fast SYBR Green Master Mix (Applied Biosystems, #4385610). qPCR results were analyzed by comparative CT (ddCT) method using *GAPDH* as an internal control. The following qPCR primers were used (5'–3'): *hMETTL2A/2B* qPCR (F): GAAGCCGGTTCCT-GAGAG; *hMETTL2A/2B* qPCR (R): ACTTGTTTCTCCTGGCACAC; *hMETTL6* qPCR (F): AGTGTTACCGTCAGTTTCAGAG; *hMETTL6* qPCR (R): GCCTCTTGTTCCAATTTCTGC; *hATM* qPCR (F): ATTCC-GACTTTGTTCCCTCTG; *hATM* qPCR (R): CATCTTGGTCCCCAT

TCTAGC; *hCDC25C* qPCR (F): CCACTCAGCTTACCACTTCTG; *hCDC25C* qPCR (R): ACCATTCGGAGTGCTACAAAG; hCHK1 qPCR (F): TTGTGGAAGACTGGGACTTG; hCHK1 qPCR (R): ATTTTCTGGA-CAGTCTACGGC; *hGAPDH* qPCR (F): CCACATCGCTCAGACACCAT; *hGAPDH* qPCR (R): CCAGGCGCCCAATACG.

## Small RNA isolation and Northern blot

Small RNA (<200nt) was isolated by using the miRVana miRNA Isolation Kit (Invitrogen, #AM1561) following the manufacturer's protocol. For Northern blot analysis of RNA, 100–1000 ng of total or small RNA were mixed with 2xTBE-Urea loading buffer (Invitrogen, #LC6876), denatured at 95 °C for 5 min, and placed on ice before loading. RNA samples were then separated by 15% TBE-Urea gels (Bio-Rad Labor7a-tory, #3450092) in 1xTBE running buffer at 200 V for 1 h. Then RNAs were transferred onto nylon membrane in 0.5xTBE transfer buffer at 30 V for 2 h, UV-crosslinked twice at 2400 × 100 μJ/cm$^2$ energy, pre-hybridized, and blotted with p32-radioactive probes against indicated tRNAs. 7 was used as a loading control. The following probes were used (5′–3′): tRNA-Arg-GCT: CACCCCAGATGGGACTCGAA; tRNA-Thr-AGT: TCGAACCCAGGATCTCCTGT; tRNA-Ser-GCT: TGGGATTCGAACC-CACGCGT; tRNA-Gly: CATTGGCCGGGAATTGAACCCG; U6: TGGAAC GCTTCACGAATTTG.

## Protein extraction and western blot

Cells were lysed using 1x Passive Lysis Buffer (Promega, #E1941) at room temperature for 15 min. Cell lysates were cleared by centrifuge at top speed at 4 °C for 10 min. Protein concentration was then deter-mined by Bradford Assay. Equal amounts of protein lysates were resolved by SDS–PAGE and transferred to the PVDF membrane in a wet transfer system. Membranes were incubated with anti-Flag-M2-HRP antibody (Sigma, #A8592) at a dilution of 1: 10,000; anti-β-actin anti-body (abcam, #ab8227) at a dilution of 1: 5,000; anti-GAPDH antibody (14C10, Cell Signaling Technology, #2118), anti-ATM antibody (D2E2, Cell Signaling Technology, #2873) at a dilution of 1:1000; anti-CDC25C antibody (Proteintech, #25887-1-AP), anti-CHK1 antibody (Proteintech, #16485-1-AP) at a dilution of 1: 2,000; anti-METTL2 (Proteintech Cat No. 16983-1-AP) at a dilution of 1: 1,000; and anti-METTL6 (Proteintech Cat No. 16527-1-AP) at a dilution of 1: 1,000. HRP-conjugated secondary antibody was used. Pierce ECL substrates (Thermo-Fisher, #32106) were used for protein band visualization.

## RNA immunoprecipitation (RIP)

HEK293T cells were transfected with pCMV2-Flag-empty vector; pCMV2-Flag-METTL2A-WT or R362A; pCMV2-Flag-Flag-METTL6-WT or R259A for 48 h. After PBS wash, cells were UV-crosslinked at 4000 × 100 μJ/cm$^2$ energy. Cell pellets were scraped and lysed in 500 μl of RIP lysis buffer (50 mM Tris–HCl pH7.4, 100 mM NaCl, 1% NP-40, 0.1% SDS, 0.5% sodium deoxycholate, protease inhibitor cocktail, RNaseOut). Cell lysates were treated with 10 μl of RQ1 RNase-free Dnase (Promega, #M6101) at 37 °C for 10 min. After centrifuge, cell lysates were collected. Protein concentrations were measured by Bradford Assay. 5% of lysates were saved as input and equal amounts of protein lysates were used for RIP. Flag-tagged proteins and their binding partners were immunoprecipitated by using the ANTI-FLAG® M2 Affinity Gel (Millipore Sigma, #A2220). The beads were washed twice with CLIP wash buffer (1×PBS, 0.1% SDS, 0.5% NP-40) and twice with high slat wash buffer (5×PBS, 0.1% SDS, 0.5% NP-40). After the last wash, the beads were resuspended in 150 μl of RIP lysis buffer. 5–10 μl of RIP samples were saved for protein analysis by western blot. 5 μl of Proteinase K and 3 μl of RNaseOut were added to the rest of the RIP samples at 42 °C for 1 h. Then RNA was extracted by adding 1 ml of TRIzol™ LS reagent (Invitrogen, #10296028) following the manu-facturer's protocol. Equal volumes of RIP samples with different plas-mid transfections were analyzed by Northern blot to detect indicated tRNAs.

## tRNA aminoacylation assay

tRNA aminoacylation was analyzed by acid urea PAGE followed by Northern blots. For uncharged tRNA controls, total RNA was treated with 0.2 M Tris-HCl pH 9.5 with 1 mM EDTA. For charged tRNA samples, total RNA was maintained in an AcE storage buffer (8 M urea, 10 mM sodium acetate pH 5.0, 1mM EDTA). Then both charged and uncharged tRNA samples were resolved by 6.5% acid urea PAGE gels (pH 5.0) using cold acid running buffer (100 mM sodium acetate pH 5.0, 1 mM EDTA). After electrophoresis, RNAs were transferred to transferred onto a nylon membrane, UV-crosslinked, pre-hybridized, and blotted with p32-radioactive probes against indicated tRNAs.

## Recombinant AlkB purification

pET30a-AlkB-WT (Addgene, #79050) and pET30a-AlkB-D135S (Addgene, #79051) plasmids were transfected into One Shot® BL21 Star™ (DE3) chemically competent cells (Invitrogen, #C601003). Single clones were cultured in a 5 ml LB medium containing 50 μg/ml Kanamycin at 37 °C overnight. On the next day, bacteria were inoculated and cultured in 250 ml LB medium containing 50 μg/ml Kanamycin at 37 °C until OD$_{600}$ reached 0.5. Protein expression was induced by adding 0.5 mM IPTG (Invitrogen, #AM9464) at 16 °C overnight. Cells were lysed by sonica-tion. Recombinant proteins were purified using Ni-NTA Agarose (Qiagen, #30210) following the manufacturer's protocol. Recombinant AlkB-WT and AlkB-D135S proteins were analyzed by SDS-PAGE followed by col-loidal blue staining (Invitrogen, #LC6025) and dialyzed into BC100 low salt buffer (20 mM Tris−HCl pH 7.8, 10% glycerol, 100 mM KCl).

## HAC Northern blot and HAC-Seq

HAC Northern blot and HAC-Seq were performed as previously described[22]. Briefly, total RNA or isolated smRNA (<200nt) samples were treated with 25 μl of ice-cold 10% hydrazine (Millipore Sigma, #215155), with 3 M NaCl on ice and in the dark for 4 h. Then 225 μl of H$_2$O was added to dilute the hydrazine concentration and to stop the hydrazine reaction. RNAs were precipitated by adding 25 μl of sodium acetate pH 5.0 and 0.75 ml of 100% ethanol. After ethanol precipita-tion, RNA pellets were resuspended in 100 μl of aniline cleavage buffer (H$_2$O: glacial acid: aniline = 7:3:1) and incubated in the dark at room temperature for 2 h. Then RNAs were purified by ethanol precipitation and dissolved in H$_2$O. RNA concentrations were measured by using Nanodrop. For Northern blot analysis, equal amounts of untreated control and HAC-treated RNA samples were separated by electro-phoresis using 15% urea-PAGE gels and blotted by using p32-radioactive probes targeting the 3′ end of indicated tRNAs.

For HAC-Seq, smRNA samples were used for the generation of HAC-seq libraries. All HAC-Seq datasets have two biological replicates. 1 μg of untreated control and HAC-treated smRNAs were demethylated with 100 μl of demethylation buffer containing 0.8 μM recombinant AlkB-WT, 1.6 μM recombinant AlkB-D135S proteins, and RNaseOut at room temperature in the dark for 2 h. 2× demethylation buffer con-tains 600 mM KCl, 4 mM MgCl$_2$, 0.1 mM (NH$_4$)$_2$Fe(SO$_4$)$_2$.6H$_2$O, 0.6 mM 2-ketoglutarate, 4 mM L-ascorbic acid, 0.1 mg/ml BSA, and 0.1 M MES, pH 5.0. The demethylation reaction was quenched with 5 mM EDTA. RNAs were purified by acid phenol-chloroform before library pre-paration. Lastly, 100 ng of demethylated control and HAC-treated smRNA samples were converted to cDNA libraries by using the NEB-Next Small RNA Library Prep Set for Illumina (NEB, #E7330). To increase RT efficiency, RNAs were incubated at 60 °C for 1 h in the reverse transcription step. Libraries were sequenced using Illumina Nextseq 500. HAC-Seq data analysis was performed as reported before[22]. After adapter trimming and quality control, clean reads were mapped to the mature tRNA sequences downloaded from GtRNAdb using Bowtie. A maximum of two mismatches were allowed. m$^3$C modification sites were determined by calculating the cleavage ratio at each single nucleotide on tRNA. Cleavage ratio at site$_i$ was calculated as the ratio of the number of reads starting at site$_{i+1}$ to the read depth of

site$_{i+1}$. The average cleavage ratios at all tRNA m$^3$C modification sites from two biological replicates were compared among WT, KO, and different enzymatic rescue groups. HAC-induced cleavages of tRNA were further visualized on IGV[51]. The untreated control samples were used for tRNA expression analysis following the ARM-seq data analysis pipeline[52]. tRNAs with a 1.5-fold difference and FDR < 0.05 were considered as differentially expressed tRNAs.

### Ribosomal footprinting (Ribo-Seq)

Ribo-Seq was performed following the TruSeq Ribo Profile (Mammalian) Library Prep Guide (Illumina) with several modifications. HEK293T WT and M2,6KO cells were grown at 80 to 90% confluency in 15-cm dishes. Cells were briefly washed with cold PBS and scraped. After centrifuge, cell pellets were flash-frozen in liquid nitrogen. Cells were then lysed in 800 μl of lysis buffer containing cycloheximide (CHX) (Millipore Sigma, #C1988) on ice for 10 min. Lysis buffer contains 20 mM Tris–HCl pH 7.4, 150 mM NaCl, 5 mM MgCl$_2$, 100 μg/ml CHX, 1% Triton X-100, 1 mM DTT, 10 U/ml RQ1 RNase-free Dnase (Promega, #M6101), 0.1% NP-40. Lysates were cleared by centrifuge. 5 A260 units of lysates were digested with 500 units of RNase I (Invitrogen, #AM2294) at room temperature with rotation for 45 min to generate ribosome-protected fragments (RPFs). RNase I reaction was stopped by adding 30 μl of SUPERase•In™ (Invitrogen, #AM2694). The digested RPFs were purified by using MicroSpin S-400 columns (Cytiva, #27514001). 2.5 μg of purified RPFs were next subjected to ribosomal RNA (rRNA) removal using riboPOOL human ribo-seq oligos (siTOOLs Biotech, #dp-P012-000042) and Dynabeads™ MyOne™ Streptavidin C1 beads (Invitrogen, #65001) following the manufacturer's protocol. rRNA-depleted RPFs were resolved on 15% TBE-Urea PAGE gels. RPFs corresponding to 25–35 nt were gel-excised and purified. In parallel, 1 A260 units of lysates were saved as RNA input samples and extracted. 1 μg of total RNA inputs were subjected to rRNA removal using RiboMinus™ Eukaryote Kit v2 (Invitrogen, #A15020), fragmented by incubating in T4 PNK buffer (NEB, #B0201) at 94 °C for 25 min. Both RPFs and fragmented total RNA inputs were end-repaired by T4 PNK (NEB, #B0201) and further purified by using RNA Clean & Concentrator-5 (Zymo Research). Lastly, both RPFs and total RNA input samples were converted into cDNA libraries by using the NEB-Next Small RNA Library Prep Set for Illumina (NEB, #E7330). cDNA libraries were further purified using MinElute PCR purification kit (Qiagen, #28004) and size-selected by electrophoresis on 6% TBE gels. Libraries were sequenced with Illumina Novaseq 6000. Ribo-seq data analysis was conducted using RiboToolkit (http://rnainformatics.org.cn/RiboToolkit/)[53]. The cleaned RPFs were collapsed into FASTA format and uploaded on RiboToolkit. Together with the gene read counts from RNA input samples, codon occupancy, ribosome pausing, and translation efficiency (TE) were compared between WT and KO groups. Genes with a 1.5-fold difference in TE and FDR < 0.05 were considered as differential translated. Gene Ontology enrichment of differentially translated genes was analyzed by RiboToolkit as well.

### HPLC-MS/MS analysis of RNA

100–1000 ng of total RNA or isolated small RNA (<200 nt) was digested with 100 U of S1 nuclease or P1 nuclease (Thermo-Fisher, #EN0321, NEB M0660S) at 37 °C for 2 h and dephosphorylated with 1 U rSAP (NEB, #M0371S). 100 μl digested samples were then filtered with Millex-GV 0.22u filters (Millipore Sigma, #SLGV033RS). 5–10 μl from each sample was injected into Agilent 6470 Triple Quad LC/MS instrument with Agilent Zorbax Eclipse C18 reverse phase HPLC column. Samples were run at 500 μl/min flow rate in mobile phase buffer A (water with 0.1% formic acid) and 0–20% gradient of buffer B (acetonitrile with 0.1% formic acid). MRM transitions are measured for cytidine (244.1–112.1), 5-methylcytidine and 3-methylcytidine (m$^5$C and m$^3$C) (258.1–126.1). m$^5$C (Cayman Chemical, #16111) and m$^3$C (Cayman Chemical, #21064) standards were run on HPLC/MS–MS to optimize

the HPLC method and determine retention times for each nucleoside. Agilent Mass Hunter LC/MS Data Acquisition Version B.08.00 and quantitative analysis version B.07.01 software was used for LC/MS–MS data collection and analysis.

### Luciferase reporter assay

Six repetitive codons were inserted after the start codon ATG of the firefly luciferase gene in the psicheck2 vector by using the Q5 site-directed mutagenesis kit (NEB, #E0554). The luciferase reporters were then transfected into cells by using Lipofectamine 2000 (Thermo-Fisher, #11668019). After 48 h transfection, firefly and renilla luciferase activities were measured by using the Dual-Glo Luciferase Assay System (Promega, #E2940). Relative firefly luciferase activity was normalized by the renilla luciferase activity and compared between different groups.

### Cell proliferation, cell cycle, and cell viability analysis

Cell proliferation was measured by seeding $1.0 \times 10^4$ cells at day 0 followed by cell-number counting on day 2, day 4, and day 6. Cell cycle phases were determined by DNA content staining using propidium iodide (PI) followed by flow cytometry analysis. Cells were prepared in single-cell suspension and fixed with ice-cold 70% ethanol at 4 °C for at least 1 h. Then, samples were treated with 100 μg/ml RNase A (Thermo-Fisher, #EN0531) at 37 °C for 30 min followed by incubation with 50 μg/ml PI (Invitrogen, #P3566) at room temperature for 5 min. DNA content was analyzed by flow cytometry on BDFortessa LSRII Cell Analyzer (BD Biosciences). At least 300,000 events were collected for each sample and data was analyzed by using FlowJo V10. Cell viability was analyzed by using CellTiter-Glo 2.0 Cell Viability Assay (Promega, #G92242). Equal number of cells were cultured with various concentrations of Cisplatin (Selleckchem, #s1166) for 48 h in a 96-well plate, and luminescent signals were measured following the manufacturer's protocol. Luminescent signals from untreated cells were used to calculate cell viability.

### Statistics and reproducibility

Detailed statistical analysis methods and sample numbers ($n$) were described in individual figure legends. GraphPad Prism and Microsoft Excel were used for data presentation. Paired or unpaired two-tailed Student $t$-tests, Fisher's exact tests, or Mann–Whitney two-sided $U$ tests were used for two comparisons. Statistical significance is considered for all analyses where *$p < 0.05$, **$p < 0.01$, ***$p < 0.001$, ****$p < 0.0001$.

### Reporting summary

Further information on research design is available in the Nature Portfolio Reporting Summary linked to this article.

## Data availability

The data supporting the findings of this study are available from the corresponding authors upon request. High-throughput sequencing data have been deposited in the Gene Expression Omnibus (GEO) under the accession numbers GSE223418 (Ribo-Seq) and GSE223469 (HAC-Seq). Source data are provided with this paper.

## Code availability

The software and algorithms for data analyses used in this study are all well-established from previous work and are referenced throughout the manuscript.

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

## Acknowledgements

R.I.G. was supported by an Outstanding Investigator Award (R35CA232115) from the National Cancer Institute (NCI) of the NIH. We thank Xin Yang (Boston Children's Hospital) for providing the pCMV2-Flag-METTL6 vector. We also thank Ronald Mathieu from Boston Children's Hospital Flow Cytometry Research Core for his help with the cell cycle analysis.

## Author contributions

J.C., E.S., and R.I.G. designed the research. J.C., E.S., S.K., and J.Y.F. performed the experiments. Q.L. performed all of the bioinformatics analysis. E.S. performed all the HPLC–MS/MS analysis of RNA. J.C., E.S., and R.I.G. analyzed the data and wrote the paper with input from other authors.

## Competing interests

R.I.G. is a co-founder, scientific advisory board member, and equity holder of Redona Therapeutics. The Gregory lab receives or has received research funding from Sanofi, Astellas, and Ono. All other authors declare no competing interests.
