## [Peer Review File · Nature Communications]

m3C32 tRNA modification controls serine codon-biased mRNA translation, cell cycle, and DNA-damage responseREVIEWER COMMENTS

Reviewer #1 (Remarks to the Author):

RNA posttranscriptional modifications play essential roles in RNA structure and function. Among all modifications, most occur in transfer RNAs (tRNAs), in particular in the anticodon loop domain. m³C at C32 (m³C32) is a modification found only in eukaryotic tRNAs. Interestingly, one budding yeast *Trm140* gene is for modification of cytoplasmic tRNA^{Thr} and tRNA^{Ser} while about four genes (*METTL2A*, *2B*, *6* & *8*) are evolved for m³C32 modification at a wider range of tRNAs. Despite identification of these enzymes in lower and higher eukaryotes, function of m³C32 remains largely unknown except its contribution to mitochondrial translation obtained by *METTL8*-deleted cells. Thus, investigation of cellular function of tRNA m³C32 modification is of significance in the field.

In this manuscript, Cui et al deleted *METTL2A/2B* or *METTL6* or combined deleted *METTL2A/2B/6* to investigate the role m³C32 on cellular functions. They first mapped the m³C32 site (despite already well established by others) using HAC-seq and then performed Ribo-seq to find decoding efficiency of codons and altered pathways. The main conclusions include tRNA^{Ser}GCT was potentially jointly modified by three enzymes and AGT codon after loss of *METTL2A/2B/6* was translated more slowly and cell cycle and DNA repair pathways were influenced. However, the main evidences supporting these two conclusions were largely insufficient and sometimes illogic and this manuscript suffers from several obvious weakness as follows.

Major points:

- 1, In construction of all deleted cell lines, only PCR pictures with poor resolution (Figure S1b) and RT-qPCR (Figure S1c) were shown; most important immunoblotting data showing loss of proteins were all missing. This WB data should be very important to assess whether they were still residual proteins in the cells. In this regard, this reviewer thinks that the cleavage ratio of tRNA^{Ser}GCT in M6KO cells is probably due to residual *METTL6* other than catalysis by *METTL2A* and *2B*.
- 2, In the abstract, "we define *METTL2A/2B/6*-dependent methylomes". In fact, mammalian m³C methylome has been well established (PMID: 27354703; 27613580). This is not a novel finding at all to identify human m³C32 methylomes.
- 3, Abstract, Pages 5 & 7 and other paragraphs, decreased m³C32 on tRNA^{Ser}GCT isodecoders was only observed with combined *METTL2A/2B/6* deletion. Again, there is no western blot data showing total deletion of *METTL6* in M6 KO cells. If *METTL6* was not absolutely deleted, the remaining *METTL6* is able to modify tRNA^{Ser}GCT. In fact, in HAC-seq (Figure 1a), modification of tRNA^{Ser}CGA isoacceptors was also not abolished after *METTL6* knockout, again suggesting a possibility of residual *METTL6* in the M6 KO cells. Even if *METTL6* is completely lacking, the conclusion of tRNA^{Ser}GCT jointly modified by *METTL2A/2B/6* is unsolid. This conclusion/suggestion is inconsistent with all previous findings from genetic and biochemical studies (PMID: 27354703; 28655767; 32923617; 34268557) from yeast, mice and human cells. Especially in PMID: 32923617, loss of *METTL6* caused loss of m³C in tRNA^{Ser}GCT (most obvious among all tRNA^{Ser} isoacceptor) from around 1.0 to 0.5 (see Figure 4C of that work); the remaining m³C is probably from m³C47d. Moreover, in PMID: 34268557, neither tRNA^{Ser}GCT transcript nor t⁶A37-tRNA^{Ser}GCT is modified by *METTL2A/2B* in vitro. Thus, modification of tRNA^{Ser}GCU by *METTL2A/2B/6* is the biggest weakness of this work and the authors should provide more detailed and reliable both biochemical and genetic data to demonstrate *METTL2A/2B*-catalyzed m³C32 modification of tRNA^{Ser}GCT by in vitro activity reconstitution and also in vivo data.
- 4, Page 7 and page 13 lines 349-352, tRNA^{Thr} sequence difference in the D-stem was suggested to influence recognition and modification by *METTL2A/2B*. This reviewer thinks it is not rigorous to have this conclusion based on the present data, which should be augmented by additional data.
- 5, Figures 2 and S3, HAC northern blot. in the method section (lines 572, 573, page 22), it was described that "equal amounts of untreated control and HAC-treated RNA samples were separated by electrophoresis" However, variable mature tRNA^{Thr}, tRNA^{Arg} and tRNA^{Ser} were observed after HAC treatment. This is in contrast with the conclusion that "knockout of *METTL2A/2B* and/or *METTL6* in HEK293T cells did not substantially alter the expression levels of tRNAs regardless of their m³C32

modification state" obtained by smRNA-seq.

6, It was found that m3C32 played no role in tRNA aminoacylation. However, the relevant Figure S4b was also with a low resolution and lacked a quantitative analysis. Besides, the claim that loss of m3C32 impaired tRNA conformation (Figure S4c) was also unobvious. The reviewer cannot observe any change in mobility.

7, Figure S4b and S4c, the membrane for northern blot in aminoacylation or tRNA conformation analyses seems to be first incubated with one tRNA probe and then stripped for another two tRNA probes. This is not rigorous considering specificity of different probes on the same membrane. Each membrane should be prepared and detected with individual probe separately. Besides, why stained with EB in Figure S4b? It is not a good choice to serve as a loading control.

8, Cell growth, proliferation, DNA damage response and cisplatin sensitivity were only superficially touched. The readers are unable to figure out a detailed mechanism why these pathways are substantially influenced, which significantly weakens the general interest and depth of this study.

Minor points:

1, Page 4, line 99, tRNA isoacceptor replicated.

2, Page 8, rationale for selection of specific residues for mutagenesis is lacking. For example, why other conserved polar residues (R316, T320, R349 etc?) in the C-terminal part were not mutated?

3, Figures 2e, S4b, S4c, tRNA isodecoder should be specified. In Figure 2e and page 14 lines 376-377, binding of a tRNA by a given protein does not necessarily mean the tRNA is a substrate of this protein.

4, Figure S2, Figure S2a, the tRNA sequence alignment lacked sequence of tRNAThrCGT-4-1; besides, legend highlighted "Different nucleotides between tRNA-ThrCGT-1-1/-3-1 and tRNA-ThrCGT-2-1/-4-1 are highlighted in blue." However, the blue box indicated the same sequence.

5, In Ribo-seq (Figure 3C), the decoding efficiency of His, Gln, Pro, Ala codons was also slowed down without any description and interpretation.

Reviewer #3 (Remarks to the Author):

In this article, Cui et al identify the tRNA substrates of the human METTL2A/2B and METTL6 m3C tRNA methyltransferases, enzymes which have previously been partially characterized by several other groups during the past few years. Using the HAC technique developed by this laboratory, the authors of this study show that METTL2A/B enzymes are essential for the formation of m3C at position 32 of tRNA-Arg and tRNA-Thr, whereas METTL6 is modifying C32 from tRNA-Ser (mostly confirming results published by Ignatova et al; Science advances; 2020; doi : 10.1126/sciadv.aaz4551). They also show that the depletion of these three enzymes is necessary to prevent formation of m3C32 on few specific tRNA-Ser. Next, the authors identify R362 from METTL2A/B and R259 from METTL6 as essential for m3C32 formation and tRNA binding. Finally, using a specific cell line lacking all three enzymes, they show that m3C32 on tRNAs is important for the translation of proteins controlling cell cycle or involved in DNA repair but also for HEK283T cell proliferation. Whether this latter observation is due to the absence of the three enzymes or to the absence of only one of these proteins is not clear since the depletion of METTL6 in mESCs also significantly affects their proliferation (Ignatova et al; Science advances; 2020; doi : 10.1126/sciadv.aaz4551). As the different cell lines are available, it could be nice to check whether the M2A/B KO and M6 KO cell lines have growth defects compared to the M2,6 KO cell line.

Overall, this work has been well-performed and the conclusions are supported by the experiments. This study brings a nice set of data that offer a deeper understanding of the roles of these three m3C32 tRNA methyltransferases.

Few points to be addressed:

As indicated above, it could be nice to check whether the M2A/B KO and M6 KO cell lines have growth defects compared to the M2,6 KO cell line. Since the cell lines have been generated, this should not be

too difficult to do.

age 10, lane 255. It is stated that « tRNA without m3C32 migrated slightly slower than m3C32-modified tRNA ». This is clear for tRNA-Arg and tRNA-Thr in the METTL2A/B KO cell lines but it is less obvious for tRNA-Ser in the M2,6 KO cell line (Supplementary figure 4C). It should be clarified.

Figure 1b and 1c : The meaning of the position highlighted in orange should be explained in the legend for this figure.

Figure 5 : The order of the panels and their description in the legend (page 33) are different. This should be corrected.

Supplementary figure 2 : There is a problem with the alignment provided. It is not readable. This should be fixed.

Typos :

Page 13, lane 347 : Should be « dependent », not « depepent ».

Page 16, lanes 416-417 : The sentence « The role of aberrant of m3C tRNA modification in human diseases is not well understood » should be clarified.

Page 31, lane 823 : 293T should be HEK293T. same is true page 34, lane 892.

Rebuttal Letter

We are very grateful to all 3 reviewers for their overall enthusiasm, constructive feedback, and helpful suggestions, and the Editor for their continued interest in our work. We are pleased to now submit a revised version of our manuscript 'm³C32 tRNA modification controls serine codon-biased mRNA translation, cell cycle, and DNA-damage response' that addresses all the reviewers' comments as described in detail below.

Reviewer #1 (Remarks to the Author):

RNA posttranscriptional modifications play essential roles in RNA structure and function. Among all modifications, most occur in transfer RNAs (tRNAs), in particular in the anticodon loop domain. m³C at C32 (m³C32) is a modification found only in eukaryotic tRNAs. Interestingly, one budding yeast Trm140 gene is for modification of cytoplasmic tRNA^{Thr} and tRNA^{Ser} while about four genes (METTL2A, 2B, 6 & 8) are evolved for m³C32 modification at a wider range of tRNAs. Despite identification of these enzymes in lower and higher eukaryotes, function of m³C32 remains largely unknown except its contribution to mitochondrial translation obtained by METTL8-deleted cells. Thus, investigation of cellular function of tRNA m³C32 modification is of significance in the field.

In this manuscript, Cui et al deleted METTL2A/2B or METTL6 or combined deleted METTL2A/2B/6 to investigate the role m³C32 on cellular functions. They first mapped the m³C32 site (despite already well established by others) using HAC-seq and then performed Ribo-seq to find decoding efficiency of codons and altered pathways. The main conclusions include tRNA^{Ser}GCT was potentially jointly modified by three enzymes and AGT codon after loss of METTL2A/2B/6 was translated more slowly and cell cycle and DNA repair pathways were influenced. However, the main evidences supporting these two conclusions were largely insufficient and sometimes illogic and this manuscript suffers from several obvious weakness as follows.

We are very grateful for the reviewer's interest in our work on the molecular and cellular function of m³C32 modification of cyto-tRNAs and their useful suggestions for further strengthening our original and timely conclusions.

Major points:

1, In construction of all deleted cell lines, only PCR pictures with poor resolution (Figure S1b) and RT-qPCR (Figure S1c) were shown; most important immunoblotting data showing loss of proteins were all missing. This WB data should be very important to assess whether they were still residual proteins in the cells. In this regard, this reviewer thinks that the cleavage ratio of tRNA^{Ser}GCT in M6KO cells is probably due to residual METTL6 other than catalysis by METTL2A and 2B.

We acknowledge the reviewer's important point. To thoroughly address this, we performed extensive immunoblotting using multiple commercially available METTL2 and METTL6 antibodies and finally found antibodies and conditions that work to specifically detect either METTL2 or METTL6 proteins by Western blot. This then allowed us to further verify the METTL2, METTL6, and METTL2+6 knockout HEK293T cell lines by western blots (to further support the previously provided genotyping presented in Supplementary Figure 1b,c). The revised Supplementary Figure 1 now includes the immunoblots verifying the KO status of the cell lines (new Figure S1d and for convenience please see below) with no detectable residual protein detected by Western blot. These new Western blots further confirm our previous Sanger sequencing data showing KO of METTL2A/2B and/or METTL6. Additionally, we designed the sgRNAs targeting the first coding exon of either METTL2A/2B or METTL6 to ensure their ORFs are totally disrupted in the KO cell lines.

Furthermore, we verified these antibodies in another cell line, MCF7 cells, where we either knocked down (siRNA) or knocked out (CRISPR/Cas9) METTL2 or METTL6 and observed expected changes in protein levels. These new results are presented in the **new Supplementary Figure 6** (and for convenience please see below). Overall, these new results present strong confirmation of the knockout status of the cell lines used in this study.

2, In the abstract, “we define METTL2A/2B/6-dependent methylomes”. In fact, mammalian m³C methylome has been well established (PMID: 27354703; 27613580). This is not a novel finding at all to identify human m³C32 methylomes.

We agree that m³C methylome has been well established using different methods by others and also by our group (PMID: 33313824). However, the MTase-dependent m³C methylome has not been well characterized. Here we knocked out METTL2A/B and/or METTL6 in human cells and report the first comprehensive METTL2A/2B/6-dependent m³C methylomes. Importantly, we describe unanticipated functional redundancy between these enzymes for the tRNA-Ser-GCT isodecoder family, and thereby uncover novel regulation of the m³C methylome in cells. We therefore consider that “we define METTL2A/2B/6-dependent methylomes” to be appropriate and accurately reflects the new data we present in Figure 1.

3, Abstract, Pages 5 & 7 and other paragraphs, decreased m³C32 on tRNASerGCT isodecoders was only observed with combined METTL2A/2B/6 deletion. Again, there is no western blot data showing total deletion of METTL6 in M6 KO cells. If METTL6 was not absolutely deleted, the remaining METTL6 is able to modify tRNASerGCT. In fact, in HAC-seq (Figure 1a), modification of tRNASerCGA isoacceptors was also not abolished after METTL6 knockout, again suggesting a possibility of residual METTL6 in the M6 KO cells. Even if METTL6 is completely lacking, the conclusion of tRNASerGCT jointly modified by METTL2A/2B/6 is unsolid. This conclusion/suggestion is inconsistent with all previous findings from genetic and biochemical studies (PMID: 27354703; 28655767; 32923617; 34268557) from yeast, mice and human cells. Especially in PMID: 32923617, loss of METTL6 caused loss of m³C in tRNASerGCT (most obvious among all tRNASer isoacceptor) from around 1.0 to 0.5

(see Figure 4C of that work); the remaining m³C is probably from m³C47d. Moreover, in PMID: 34268557, neither tRNA^{Ser}GCT transcript nor t⁶A37-tRNA^{Ser}GCT is modified by METTL2A/2B in vitro. Thus, modification of tRNA^{Ser}GCU by METTL2A/2B/6 is the biggest weakness of this work and the authors should provide more detailed and reliable both biochemical and genetic data to demonstrate METTL2A/2B-catalyzed m³C32 modification of tRNA^{Ser}GCT by in vitro activity reconstitution and also in vivo data.

*We agree with the reviewer that our findings on some of the tRNA targets of these enzymes are unexpected and we consider that this adds originality to the results described in our manuscript. We thank the reviewer for their suggestions. As described above, to address the possible issue of residual METTL6 protein in the METTL6 KO cells, we have now carried out western blots using METTL2 and METTL6 antibodies. No METTL2 protein is detected in the METTL2 or METTL2+6 KO cell lines. Likewise, no METTL6 protein is detected in METTL6 or METTL2+6 KO cell lines (**new Supplementary Figure 1d**). This is consistent with the genotyping where we previously found by Sanger sequencing the complete KO of METTL2 and/or METTL6 (Supplementary Figure 1a-c) in our genotype-verified cell lines, we observe a complete loss of METTL2 or METTL6 proteins by western blot. We further generated new METTL2 and METTL6 knockout MCF7 cell lines employing different guide RNAs and observed complete loss of corresponding western blot bands for METTL2 and METTL6 proteins (**new Supplementary Figure 5**). Taken together, we are confident that the cell lines we use are knockout cell lines for METTL2 and/or METTL6. This additional verification addresses the reviewers concern about possible residual METTL6 activity in the METTL6 KO cells. Moreover, these results further support our conclusion based on HAC-Seq data that METTL2 plays an unexpected role in the m³C32 modification of the tRNA-Ser-GCT isodecoder family.*

*Our HAC-seq results suggest that m³C32 methylation of tRNA-Ser-GCT depends on both METTL2 and METTL6 enzymes. To verify the in vivo targets of these enzymes by an independent method, we carried out HPLC-MS/MS analysis of specific tRNAs isolated from WT and knockout cells. Consistent with HAC-seq data, we observed a complete loss of m³C on tRNA-Thr-AGT upon METTL2A/2B deletion, whereas loss of METTL6 did not impact m³C on this tRNA, verifying specific in vivo activity of these enzymes (**new Figure 2a**, and for convenience please see below). Strikingly, deletion of either METTL2A/2B or METTL6 both decreased the m³C level on tRNA-Ser-GCT, and the combined loss of all three enzymes resulted a greater decrease in m³C level, indicating that METTL2A/2B and METTL6 are both required for m³C modification of this tRNA (**new Figure 2a**). Also of note, the residual m³C detected by mass spectrometry on tRNA-Ser-GCT in the M2,6 KO cells is likely due to the modification of C47d residue that is catalyzed by an unknown MTase (Figure 1a, and Figure 2a). Taken together, these HAC-seq and mass spectrometry results strongly support that while METTL2A/B and METTL6 enzymes generally display distinct specificities towards either tRNA-arginine and tRNA-threonine members, or the tRNA-serine family, respectively, these MTases (METTL2A/2B/6) are redundantly involved in the m³C32 modification of tRNA-Ser-GCT isodecoders.*

Moreover, although individual loss of METTL2 or METTL6 does not have a significant impact on cell growth in HEK293T cells, combined knockout of these enzymes impacts cell growth substantially (**New Figure 5A** and for convenience please see below). Additionally, we observe a similar cell growth phenotype in MCF7 cells (**New Supplementary Figure 5**). These phenotypic changes suggest a redundancy of these enzymes at physiological level in these cell types.

Overall, our HAC-Seq, new HPLC-MS/MS analysis, and additional phenotypic analysis helped us uncover this novel redundancy between METTL2 and METTL6 enzymes both at the molecular and cellular level.

We thank the reviewer for the references to the previous publications about m^3C tRNA modification in different organisms. These previous findings are the foundation of our research. We acknowledge that while most of our findings agree well with expected roles for METTL2 and METTL6 in shaping the m^3C tRNA methylome (i.e. METTL2 modifies tRNA-Arg and tRNA-Thr at position C32, and METTL6 modifies tRNA-Ser) that there are some important differences between our new results and current literature on the functional redundancy between METTL2 and METTL6 for the tRNA-Ser-GCT family. Compared with other methods, HAC-Seq is a powerful tool to precisely study METTL2A/2B- and METTL6-mediated m^3C modification of different tRNA isodecoders and isoacceptors and can therefore comprehensively reveal METTL2-, METTL6-, and METTL2+6-dependent m^3C profiles that might have been missed in previous work. Moreover, we now add new Mass Spectrometry data (described above) that strongly supports the redundancy between METTL2 and METTL6 for modification of tRNA-Ser-GCT. Some other considerations related to the reviewer's citations include: PMID 27354703 is focused on m^3C modification in the fission yeast, and although *S. pombe* *trm140+* and *trm141+* methylate m^3C32

position at tRNA-Ser and tRNA-Thr, respectively, the m³C landscape is more expansive and its regulation more complex in human cells. For example, there is no m³C47 in *S. pombe*. It is therefore not unexpected for there to be differences across these evolutionarily divergent organisms. PMID 34268557 performed a comprehensive characterization of the *in vitro* substrate preferences of METTL2A and METTL6 using reconstituted methylation assays. While the authors did not find tRNA-Ser-GCT to be a substrate for METTL2 enzyme in these biochemical assays, this does not exclude the possibility that the situation *in vivo* is more complex and perhaps additional co-factors and/or tRNA modifications can alter the substrate determinants *in vivo* as suggested by our new (HAC-Seq, and HPLC-MS/MS spec, and further supported by Northern blot analysis of immunopurified METTL2/6 complexes from cells) findings for tRNA-Ser-GCT in human cells. PMID 32923617 used NAIL-MS to quantify m³C modification levels on purified tRNA from cells and PMID 28655767 used primer extension assays to map tRNA m³C modification sites in cells. NAIL-MS and primer extension assays are highly dependent on the specificity of the probes targeting tRNA of interests. However, due to the high sequence similarity of multiple tRNAs, it is very challenging to specifically detect pure tRNA-Ser-GCT using these approaches. Moreover, NAIL-MS cannot determine the m³C modification sites on tRNA. HAC-seq can precisely map the m³C methylome at single-nucleotide resolution throughout the transcriptome. Nevertheless, as noted by the reviewer Ignatova et al., showed by NAIL-MS ~50% reduction in m³C on isolated Ser-tRNA-GCT (Figure 4c in PMID 32923617). These results are highly consistent with our new Mass Spec data where we also see a similar level of m³C reduction on isolated tRNA-Ser-GCT (New Figure 2A and see below). We do not dispute that this tRNA is a substrate of METTL6. The novelty of our findings is that tRNA-Ser-GCT is also modified by METTL2 in cells as shown by HAC-Seq (Figure 1) and supported by mass spec where we see decreased m³C levels on tRNA-Ser-GCT isolated from METTL2 KO cells, as well as a stronger decrease on tRNA-Ser-GCT isolated from METTL2+6 KO cells compared with either METTL2 or METTL6 single KO cells (please see below highlighted in red box). Indeed, the relative contribution of METTL2 and METTL6 to the m³C32 modification of tRNA-Ser-GCT might vary in different cell lines due to the relative expression/activity of these different MTases. Our work was performed in HEK293T cells whereas other studies used different cell types (for example HAP1 cells in Ignatova et al., 2020). Thus, our new data help refine our current understanding of METTL2- and METTL6- dependent m³C methylomes. We have added sentences to the discussion of the revised manuscript to help reconcile our findings with published work as described above.

4, Page 7 and page 13 lines 349-352, tRNA^{Thr} sequence difference in the D-stem was suggested to influence recognition and modification by METTL2A/2B. This reviewer thinks it is not rigorous to have this conclusion based on the present data, which should be augmented by additional data.

We agree with the reviewer that our statement about the sequence difference in the tRNA-Thr D arm is only a hypothesis to explain the observed HAC-Seq results. Considering the reviewer's comment and since this point is not a major focus of this study, we have removed this speculation and the associated supplementary figure from the revised manuscript.

5, Figures 2 and S3, HAC northern blot. in the method section (lines 572, 573, page 22), it was described that "equal amounts of untreated control and HAC-treated RNA samples were separated by electrophoresis" However, variable mature tRNAThr, tRNAArg and tRNASer were observed after HAC treatment. This is in contrast with the conclusion that "knockout of METTL2A/2B and/or METTL6 in HEK293T cells did not substantially alter the expression levels of tRNAs regardless of their m3C32 modification state" obtained by smRNA-seq.

We have now added U6 Northern blots to these figures (now revised Figure 2b, 2c, and Supplementary Fig. 2b) as a loading control. Based on these results, as well as many other similar data (not shown) we can conclude with confidence that that loss of METTL2 and or METTL6 does not alter tRNA levels. We acknowledge that the relative tRNA levels can appear more variable after HAC treatment but this adds unnecessary complexity (i.e. after HAC treatment, the tRNAs are cleaved based on their m3C methylation status impacting their apparent levels) that can confound any interpretations related to tRNA levels that are more accurately represented in the untreated RNA samples.

6, It was found that m3C32 played no role in tRNA aminoacylation. However, the relevant Figure S4b was also with a low resolution and lacked a quantitative analysis. Besides, the claim that loss of m3C32 impaired tRNA conformation (Figure S4c) was also unobvious. The reviewer cannot observe any change in mobility.

We thank the reviewer for this suggestion. We repeated the experiments in Figure S4b and replaced with higher resolution Northern blots and clearer data (New Supplementary Figure 3b and below for convenience). The result clearly shows that METTL2 or METTL6 do not influence tRNA expression or aminoacylation. We speculate that m³C32 impacts tRNA conformation to influence tRNA function in mRNA translation (that we measure by Ribo-Seq). Indeed, the differential tRNA migration we observed on non-denaturing agarose gels provides some support of this. However, we agree with the reviewer that the evidence supporting this is quite weak and since this point is not a major focus of our current study we have removed these data and will deploy more direct and sophisticated methods in future work aimed at elucidating the impact of m³C on tRNA structure/function.

7, Figure S4b and S4c, the membrane for northern blot in aminoacylation or tRNA conformation analyses seems to be first incubated with one tRNA probe and then stripped for another two tRNA

probes. This is not rigorous considering specificity of different probes on the same membrane. Each membrane should be prepared and detected with individual probe separately. Besides, why stained with EB in Figure S4b? It is not a good choice to serve as a loading control.

We thank the reviewer for this suggestion. We repeated this experiment and replaced Supplementary Figure S4b that now includes a more appropriate U6 Northern blot loading control (See above point #6). The result clearly shows that METTL2 or METTL6 do not influence tRNA expression or aminoacylation. We speculate that m^3C32 impacts tRNA conformation to influence tRNA function in mRNA translation (that we measure by Ribo-Seq). Indeed, the differential tRNA migration we observed on non-denaturing agarose gels provides some support of this. However, since this point is not a major focus of our current study, we have removed these data and will deploy more direct and sophisticated methods in future work aimed at elucidating the impact of m^3C on tRNA structure/function.

8, Cell growth, proliferation, DNA damage response and cisplatin sensitivity were only superficially touched. The readers are unable to figure out a detailed mechanism why these pathways are substantially influenced, which significantly weakens the general interest and depth of this study.

To address this, we have performed additional experiments to expand upon the results the reviewer pointed out. We add new data from cell growth assays in HEK293T cells that now include growth curves for the WT, METTL2A/B KO, METTL6 KO, and METTL2A/B/6 KO cells (New Figure 5a and below). Similarly, we repeated and expanded the Cisplatin Sensitivity assays to now include the WT, METTL2A/B KO, METTL6 KO, and METTL2A/B/6 KO HEK293T cells (New Figure 5d and shown below).

a

d

We furthermore added new cell proliferation data for MCF7 human breast cancer cell line, where we observe that while individual deletion of METTL2 or METTL6 had a modest effect on growth of MCF7 human breast cancer cells, the further siRNA-mediated depletion of either METTL6 in the METTL2 KO

cells, or knockdown of METTL2 in the METTL6 KO cells strongly suppressed cell proliferation (**New supplementary Fig. 5a-c, see below**).

At the molecular level, our Ribo-seq analysis revealed that loss of METTL2/6 enzymes impacts the translation of DNA damage response and growth promoting genes thereby providing a molecular mechanistic explanation for these observed cellular phenotypes. To help emphasize the key original findings of this study we now include a model (**New Figure 6, see below**)

Minor points:

1, Page 4, line 99, tRNA isoacceptor replicated.

We have corrected this typo.

2, Page 8, rationale for selection of specific residues for mutagenesis is lacking. For example, why other conserved polar residues (R316, T320, R349 etc?) in the C-terminal part were not mutated?

We tried to mutate other conserved polar residues such as D297, R316, R349, N353. However, due to technical issues and also the very low observed expression of these mutants, we did not continue with mutations of these conserved residues.

3, Figures 2e, S4b, S4c, tRNA isodecoder should be specified. In Figure 2e and page 14 lines 376-377, binding of a tRNA by a given protein does not necessarily mean the tRNA is a substrate of this protein.

We have changed the labels to reflect the specific isodecoders. We agree with the reviewer that the binding of an enzyme does not necessarily mean that tRNA is a substrate. In addition to HAC-Seq and biochemical HPLC-MS/MS analysis, we performed binding assays as an independent verification of target validation.

4, Figure S2, Figure S2a, the tRNA sequence alignment lacked sequence of tRNA^{Thr}CGT-4-1; besides, legend highlighted "Different nucleotides between tRNA^{Thr}CGT-1-1/-3-1 and tRNA^{Thr}CGT-2-1/-4-1 are highlighted in blue." However, the blue box indicated the same sequence.

Related to Point #4 above, considering the reviewer's comment and since this point is not a major focus of this study, we have removed this speculation and the associated supplementary figure from the revised manuscript.

5, In Ribo-seq (Figure 3C), the decoding efficiency of His, Gln, Pro, Ala codons was also slowed down without any description and interpretation.

The decreased codon occupancy observed for several codons/amino acids is likely a consequence of compromised translation in the METTL2A/B/6 KO cells. We focus on Serine since in contrast to the more general decrease in ribosome protected fragments (RPFs) for several codons, we see obvious and specific accumulation of RPFs for Serine and in particular Ser-AGU codons (that are decoded by tRNA-Ser-GCT) (Figure 3c-e and Supplementary Figure 4). Moreover, Ser-AGU codons are amongst the most highly enriched in the set of genes with decreased Translation efficiency in METTL2A/B/6 KO cells (Figure. 4a). That is why we specifically further investigated the decoding effects on Serine AGU codon in Figure 4f and Figure 4g, and is highlighted in our new model in Figure 6.

Reviewer #2 (Remarks to the Author):

In this manuscript, the authors have comprehensively analyzed the three MTases METTL2A, 2B and 6. All three have been implicated in cytoplasmic m³C modification. The authors produced knock out cells for the individual (2A/B and 6) as well as a triple knock out cell line (2A/B/6) and used these cell lines to profile m³C modification on a global scale. The authors have recently developed HAC-seq, which induces a cleavage event at m³C modification sites. Using this approach, the authors find that METTL2A/B are primarily required for m³C modification of cytoplasmic tRNA-Thr and -Arg, while METTL6 is the enzyme for the cytoplasmic tRNA-Ser variants. The authors further mutated specific residues, which affect binding to the tRNA substrates. A surprising finding, however, was that Mettl2A/B also bind moderately to tRNA-Arg and -Thr and also seem to be somewhat promiscuous towards their substrates. Using ribo-seq, the authors find that mRNAs with specific Ser, Thr, or Arg codons are affected. However, Thr and Arg have the opposite effect compared to Ser, which appears to be stalled at the A-side. Most affected mRNAs are involved in cell cycle and DNA repair and indeed, triple knock out cells show cell cycle phenotypes and are more sensitive to cisplatin.

This is a clear and well-written manuscript. It addresses an interesting and relevant question and although it is already known that the three MTases produce m³C on cytoplasmic tRNAs, this is a comprehensive analysis with several so far unknown and unexpected findings. Particularly, the finding that addition m³C MTases are still operating on tRNAs is very interesting.

We are grateful to the reviewer for their interest and positive assessment of our work. We agree that identifying the unknown m³C methyltransferase(s) responsible for m³C32 modification of certain tRNA-Arg-CCT and tRNA-Thr-CGT isodecoders, as well as the enzymes responsible for C47d modification of tRNA-Ser and tRNA-Leu-CAG, and C20 on tRNA-Met-CAT is an important and interesting topic for future investigation, and highlights the complexity if the m³C tRNA methylome in human cells.

Below, I listed some issues that should be considered.

1. Figure 2: the description of the Figures is not clear. "M2KO" are cells in which METTL2A and B are knocked out. The rescues "M2-WT" and "M2-R362A" are only METTL2A? This should be clearly stated in the Figure. I guess METTL2B has been omitted because it has only a minor contribution. Nevertheless, it should also be added for a more comprehensive picture.

We thank the reviewer for their comment. To make this clearer we have modified the Figure legend for the rescue figure - (b) Northern-blot detection of HAC-induced cleavage of tRNA at m³C showing re-expression of METTL2A (M2) but not the METTL2A-R362A mutant (M2-R362A) in METTL2A/B knockout (M2KO) cells can rescue the m³C modification on tRNA-Thr and tRNA-Arg'. Considering the extremely high sequence identity between METTL2 paralogs with 98.4% identity and 99.2% similarity at the amino acid level throughout the entire 378 amino acids of each protein, and considering the robust rescue we observed with METTL2A expression (Figure 2b), we decided to restrict our rescue experiments to METTL2A.

2. Figure 3: In the ribo-seq data, the authors state that T, R and S tRNAs are dramatically affected. Many others, however, show also similar variation. This should be addressed or at least commented in the text. Furthermore, the A+1 site data shown in Figure 3C is not at all mentioned in the results section. If it is not important for the conclusion, it should probably be omitted.

The decreased codon occupancy observed for several codons/amino acids is likely a consequence of compromised translation in the METTL2A/B/6 KO cells. We focus on Serine since in contrast to the more general decrease in ribosome protected fragments (RPFs) for several codons, we see obvious and specific accumulation of RPFs for Serine and in particular Ser-AGU codons (that are decoded by tRNA-Ser-GCT) (Figure 3c-e and Supplementary Figure 4). Moreover, Ser-AGU codons are amongst the most highly enriched in the set of genes with decreased Translation efficiency in METTL2A/B/6 KO cells (Figure. 4a). That is why we specifically further investigated the decoding effects on Serine AGU codon in Figure 4f and Figure 4g, and is highlighted in our new model in Figure 6. A+1 site is mentioned in the text and serves as a control showing the expected A site changes in ribosome occupancy caused by altered tRNA function due to m³C deficiency.

3. Figure 5: the description of the sub-figures appears mixed-up in the text. Proliferation is 5a and not c? Effects are very small and I was wondering whether many other triple-knock-out cell lines would show similar effects. How specific is the effect of cisplatin? Another unrelated knock out cell line could serve as specificity control. At least the proliferation effects should be rescued with individual or all three proteins.

We thank the reviewer for these suggestions. To address these, we have performed additional experiments to expand upon the results the reviewer mentioned. We performed new cell growth assays in HEK293T cells that now include growth curves for the WT, METTL2A/B KO, METTL6 KO, and METTL2A/B/6 KO cells (New Figure 5a and below). Similarly, we repeated and expanded the Cisplatin Sensitivity assays to now include the WT, METTL2A/B KO, METTL6 KO, and METTL2A/B/6 KO HEK293T cells (New Figure 5d and shown below).

a

d

We furthermore added new cell proliferation data for MCF7 human breast cancer cell line, where we observe that while individual deletion of METTL2 or METTL6 had a modest effect on growth of MCF7 human breast cancer cells, the further siRNA-mediated depletion of either METTL6 in the METTL2 KO cells, or knockdown of METTL2 in the METTL6 KO cells strongly suppressed cell proliferation (**New supplementary Fig. 5a-c, see below**).

At the molecular level, our Ribo-seq analysis revealed that loss of METTL2/6 enzymes impacts the translation of DNA damage response and growth promoting genes thereby providing a molecular mechanistic explanation for these observed cellular phenotypes. To help emphasize the key original findings of this study we now include a model (**New Figure 6, see below**)

Reviewer #3 (Remarks to the Author):

In this article, Cui et al identify the tRNA substrates of the human METTL2A/2B and METTL6 m³C tRNA methyltransferases, enzymes which have previously been partially characterized by several other groups during the past few years. Using the HAC technique developed by this laboratory, the authors of this study show that METTL2A/B enzymes are essential for the formation of m³C at position 32 of tRNA-Arg and tRNA-Thr, whereas METTL6 is modifying C32 from tRNA-Ser (mostly confirming results published by Ignatova et al; Science advances; 2020; doi : 10.1126/sciadv.aaz4551). They also show that the depletion of these three enzymes is necessary to prevent formation of m³C32 on few specific tRNA-Ser. Next, the authors identify R362 from METTL2A/B and R259 from METTL6 as essential for m³C32 formation and tRNA binding. Finally, using a specific cell line lacking all three enzymes, they show that m³C32 on tRNAs is important for the translation of proteins controlling cell cycle or involved in DNA repair but also for HEK283T cell proliferation. Whether this latter observation is due to the absence of the three enzymes or to the absence of only one of these proteins is not clear since the depletion of METTL6 in mESCs also significantly affects their proliferation (Ignatova et al; Science advances; 2020; doi : 10.1126/sciadv.aaz4551). As the different cell lines are available, it could be nice to check whether the M2A/B KO and M6 KO cell lines have growth defects compared to the M2,6 KO cell line.

Overall, this work has been well-performed and the conclusions are supported by the experiments. This study brings a nice set of data that offer a deeper understanding of the roles of these three m³C32 tRNA methyltransferases.

We are grateful to the reviewer for their interest and positive assessment of our work. We appreciate the reviewer's specific points that we have addressed in our revised manuscript.

Few points to be addressed:

As indicated above, it could be nice to check whether the M2A/B KO and M6 KO cell lines have growth defects compared to the M2,6 KO cell line. Since the cell lines have been generated, this should not be too difficult to do.

We have performed additional experiments as suggested to expand upon the results the reviewer mentioned. We performed new cell growth assays in HEK293T cells that now include growth curves for

the WT, METTL2A/B KO, METTL6 KO, and METTL2A/B/6 KO cells (**New Figure 5a** and below). Similarly, we repeated and expanded the Cisplatin Sensitivity assays to now include the WT, METTL2A/B KO, METTL6 KO, and METTL2A/B/6 KO HEK293T cells (**New Figure 5d** and shown below).

a

d

We furthermore added new cell proliferation data for MCF7 human breast cancer cell line, where we observe that while individual deletion of METTL2 or METTL6 had a modest effect on growth of MCF7 human breast cancer cells, the further siRNA-mediated depletion of either METTL6 in the METTL2 KO cells, or knockdown of METTL2 in the METTL6 KO cells strongly suppressed cell proliferation (**New supplementary Fig. 5a-c**, see below).

At the molecular level, our Ribo-seq analysis revealed that loss of METTL2/6 enzymes impacts the translation of DNA damage response and growth promoting genes thereby providing a mechanistic explanation for these observed cellular phenotypes. To help emphasize the key original findings of this study we now include a model (New Figure 6, see below)

Page 10, lane 255. It is stated that « tRNA without m3C32 migrated slightly slower than m3C32-modified tRNA ». This is clear for tRNA-Arg and tRNA-Thr in the METTL2A/B KO cell lines but it is less obvious for tRNA-Ser in the M2,6 KO cell line (Supplementary figure 4C). It should be clarified.

We agree with the reviewer's comment. We speculate that m³C32 impacts tRNA conformation to influence tRNA function in mRNA translation (that we measure by Ribo-Seq). Indeed, the differential tRNA migration we observed on non-denaturing agarose gels provides some support of this. However, we agree with the reviewer that the evidence supporting this is quite weak and the migration difference for certain tRNAs is very subtle. We have tried several new experiments using larger denaturing gels and running conditions but have not been able to substantially improve the resolution of this indirect assay of tRNA conformation. Considering that this point is not a major focus of our current study we have removed these data from the revised manuscript and in future work will deploy more direct and sophisticated methods aimed at elucidating the impact of m³C on tRNA structure/function.

Figure 1b and 1c : The meaning of the position highlighted in orange should be explained in the legend for this figure.

Additional information in the legend of Figure 1 has been added to clarify this 'Orange label shown in (b) and (c) represents sequencing mutation that reflects adenosine to inosine (A to I) editing at position A34'.

Figure 5 : The order of the panels and their description in the legend (page 33) are different. This should be corrected.

Figure 5 has been revised with additional experimental data and the text has been corrected accordingly.

Supplementary figure 2 : There is a problem with the alignment provided. It is not readable. This should be fixed.

This figure has been removed based on the recommendation of Reviewer #1.

Typos :

Page 13, lane 347 : Should be « dependent », not « depernent »

Corrected.

Page 16, lanes 416-417 : The sentence « The role of aberrant of m³C tRNA modification in human diseases is not well understood » should be clarified.

We have modified this part of the discussion to include the emerging evidence of METTL2A, -B, -6 dysregulation and cancer. 'Interestingly, expression of METTL2A, METTL2B, and METTL6 mRNA is elevated in most tumor types compared with corresponding normal tissues and is associated with poor prognosis⁴³. High levels of METTL2A mRNA expression is associated with shortened overall survival in breast invasive carcinoma (BRCA) patients and associated with high grade tumors⁴³. Gene Ontology analysis of the set of mRNAs enriched in TCGA datasets based on METTL2A expression showed that several cell cycle-related pathways were enriched, such as chromosome segregation, and DNA biosynthetic process. These results indicate that BRCA cells with high METTL2A expression could be active in proliferation and DNA damage response pathways⁴³.'

Page 31, lane 823 : 293T should be HEK293T. same is true page 34, lane 892.

Corrected.

REVIEWERS' COMMENTS

Reviewer #2 (Remarks to the Author):

In the revised version of their manuscript, the authors have addressed all points that I had raised on the previous version. The authors have adequately responded and therefore I am satisfied with the response to my concerns.

Reviewer #3 (Remarks to the Author):

The authors have satisfactorily addressed all my concerns.